# Antibacterial macrocyclic peptides reveal a distinct mode of BamA inhibition

Morgan E. Walker [1,5], Wei Zhu[2,5], Janine H. Peterson[3,5], Hao Wang[1], Jon Patteson[1], Aileen Soriano[2], Han Zhang[1], Todd Mayhood[2], Yan Hou[2], Samaneh Mesbahi-Vasey[1], Meigang Gu[4], John Frost[2], Jun Lu[1], Jennifer Johnston[2], Christopher Hipolito[2], Songnian Lin [2], Ronald E. Painter[1], Daniel Klein[1], Abbas Walji [1], Adam Weinglass[2], Terri M. Kelly[2], Adrian Saldanha[2], Jeffrey Schubert[1], Harris D. Bernstein [3] ✉ & Scott S. Walker [1] ✉

Outer membrane proteins (OMPs) produced by Gram-negative bacteria contain a cylindrical amphipathic β-sheet ("β-barrel") that functions as a membrane spanning domain. The assembly (folding and membrane insertion) of OMPs is mediated by the heterooligomeric β-barrel assembly machine (BAM). The central BAM subunit (BamA) is an attractive antibacterial target because its structure and cell surface localization are conserved, it catalyzes an essential reaction, and potent bactericidal compounds that inhibit its activity have been described. Here we utilize mRNA display to discover cyclic peptides that bind to *Escherichia coli* BamA with high affinity. We describe three peptides that arrest the growth of BAM deficient *E. coli* strains, inhibit OMP assembly in live cells and in vitro, and bind to unique sites within the BamA β-barrel lumen. Remarkably, we find that if the peptides are added to cultures after a slowly assembling OMP mutant binds to BamA, they accelerate its biogenesis. The data strongly suggest that the peptides trap BamA in conformations that block the initiation of OMP assembly but favor a later assembly step. Molecular dynamics simulations provide further evidence that the peptides bind stably to BamA and function by a previously undescribed mechanism.

Antibiotic resistance is a rapidly increasing healthcare threat worldwide that has been consistently highlighted by the World Health Organization[1–4]. Significantly, no new classes of antibiotics that target Gram-negative bacteria have been commercialized since the 1960s[5]. As a result, an increase in multi-drug resistant Gram-negative bacteria has emerged in hospital settings, including the ESKAPEE pathogens *Escherichia coli*, *Klebsiella pneumoniae*, *Acinetobacter baumannii*, *Pseudomonas aeruginosa*, and *Enterobacter spp*[6,7]. Gram-negative bacteria are particularly difficult to target because they have both a cytoplasmic membrane and an outer membrane (OM), an additional barrier that is not present in Gram-positive bacteria. The OM, which is essential for survival[8], contains an inner leaflet composed of phospholipids and an outer leaflet composed of a unique negatively charged glycolipid known as lipopolysaccharide (LPS) that prevents the penetration of both large (>600 daltons) and nonpolar compounds, including some traditional antibiotics that effectively target Gram-positive bacteria[9,10]. Additionally, the OM contains a high concentration of integral membrane proteins that adopt an unusual structure. Unlike most other integral membrane proteins that have one or more hydrophobic α-helical membrane-spanning segments, outer

[1]Merck & Co., Inc., West Point, PA, USA. [2]Merck & Co., Inc., Rahway, NJ, USA. [3]Genetics and Biochemistry Branch, National Institute of Diabetes and Digestive and Kidney Diseases, National Institutes of Health, Bethesda, MD 20892, USA. [4]Evotec Ltd., Abingdon, Oxfordshire OX14 4RZ, UK. [5]These authors contributed equally: Morgan E. Walker, Wei Zhu, Janine H. Peterson. ✉e-mail: harris_bernstein@nih.gov; scott.walker@merck.com

membrane proteins (OMPs) span the membrane via an amphipathic β sheet that folds into a cylindrical "β-barrel" structure[11]. The OMPs have unique potential as antibacterial targets due to their accessible location on the exterior of the cell and their specificity to Gram-negative bacteria[12]. Two highly conserved OMPs that play key roles in the formation of the OM, LptD and BamA, are of particular interest because they are essential for viability[13–15]. LptD inserts LPS molecules that are produced in the cytoplasmic membrane and then transported through the periplasm directly into the outer leaflet of the OM[13]. BamA is the central subunit of the β-barrel assembly machinery (BAM), a hetero-oligomeric complex that catalyzes the assembly (folding and membrane insertion) of OMPs[14,15].

In recent years, our understanding of the mechanism by which BAM mediates OMP assembly has advanced considerably. It is now clear that although BAM is composed of a variable number of subunits, BamA is the major functional component and is sufficient to catalyze OMP assembly under very specific conditions[16]. In addition to containing a 16-stranded β-barrel, BamA contains five periplasmic polypeptide transport-associated (POTRA) domains that in *E. coli* bind to four lipoproteins, BamB-E[15,17,18]. BamD is the only lipoprotein that is essential[19], and although there is evidence that BamD binds directly to newly synthesized OMPs, its function appears to be regulatory[16,20]. BamB, BamC, and BamE are nonessential and their exact functions are poorly understood, but they facilitate OMP folding to varying degrees[21]. Unlike the β-barrels of most OMPs, which are extremely stable because their first and last β-strands are held together by a large number of hydrogen bonds that form along the entire length of both β-strands, the first and last β-strands of BamA are held together by only four hydrogen bonds that form at the β-strand termini. As a consequence, the BamA β-barrel can both open laterally and distort the local lipid environment[22–26]. Biochemical analysis and cryo-EM structures of BAM-OMP assembly intermediate supercomplexes have revealed that during assembly the BamA β-barrel shifts from a lateral closed state to a lateral open state and thereby enables a conserved C-terminal motif in the substrate (the "β-signal") to bind tightly to its first β-strand (BamA β1)[25–29] in an antiparallel configuration. Subsequently, the substrate enters the OM and is progressively converted from a curved β-sheet to a barrel-like structure. This process is promoted by the inherent rigidity of the OM that results from the linkage of LPS molecules by divalent cations[25,30]. The substrate forms an asymmetric hybrid barrel with the open BamA β-barrel, and has been proposed to swing into the OM through weak, dynamic interactions between the N-terminal β-strand of the substrate and the C-terminal β-strands of BamA[27]. Finally, the substrate β-barrel closes and is released from BamA by a strand-exchange mechanism, and BamA returns to a lateral closed state[26]. Despite these advances, however, many aspects of the assembly process are still unclear, including the role of specific lipids, membrane distortion, the higher-order organization of the OM in OMP assembly, and the energetics of a process that occurs in an environment that lacks ATP[31–33].

Although BamA has a large flat surface and, unlike typical enzymes, lacks an active site that could be targeted for inhibition, several groups have validated the protein as an antibacterial target by discovering BamA inhibitors that show potent bactericidal activity[34,35]. All of these compounds, however, have drawbacks. Genentech developed a monoclonal antibody that selectively targets the extracellular loops of BamA, but it only kills specific mutant strains of *E. coli*[36]. Based on cellular and genetic evidence, Hart et al. described the discovery of MRL-494, a synthetic small molecule with broad-spectrum Gram-negative antibacterial activity, that acts through an unknown mechanism involving BamA. However, the compound exhibits off-target activity against Gram-positive bacteria[37]. Several other small molecules were identified from library screens based on their ability to inhibit the biogenesis of a model OMP or to induce σE and Rcs stress responses. These compounds might block the activity of BamA, but

their molecular target remains to be identified[38,39]. Polyphor AG (now Spexis AG) described a family of bactericidal chimeric-peptidomimetics that bind to the exterior of BamA as well as other OMPs and LPS, and an academic group identified a bactericidal peptidomimetic (JB-95) that binds to BamA and LptD and disrupts the OM, but the mechanism of action of these compounds is unclear[40–42]. Lastly, the darobactins, natural products first discovered in *Photorhabdus khanii*, are ribosomally synthesized peptides that are modified post-translationally to form small bicyclic ring structures with broad Gram-negative antibacterial activity[43]. The darobactins bind to the BamA β-barrel seam by forming an extensive network of backbone hydrogen bonds with BamA β1 that enables them to function as competitive inhibitors of β-signal binding, offering an explanation for their broad-spectrum activity[44]. The discovery of darobactin initiated several derivatizations through manipulation of the biosynthetic gene cluster[45–48] and natural product exploration studies[49,50], but because of the complex cyclizations in the molecule, darobactins are difficult to synthesize de novo[51,52].

The discovery and further refinement of the darobactins prompted us to initiate a hit discovery campaign to identify more synthetically tractable BamA inhibitors. Given that darobactins are based on a bicyclic peptide scaffold (and that other peptidomimetics have been shown to at least bind to BamA), we reasoned that constrained cyclic peptides might be a suitable modality for BamA inhibition. Here we describe the use of an mRNA display screening platform[53] that contains a large library of monocyclic peptides (over $10^{12}$ unique compounds) to isolate peptides that both bind to BamA with high affinity and show antibacterial activity. Cyclic peptides that meet these criteria might be ideal lead compounds for new antibiotics as they are easier to produce and allow the use of a larger unnatural amino acid palette. Indeed after we submitted this manuscript, the isolation of two peptidic BamA inhibitors in an mRNA display campaign, one that binds to the exterior of the BamA β-barrel in the laterally closed state and one that binds to the lumen of the BamA β-barrel in the laterally open state, was reported[54].

In this work, we validate the BamA binding and antibacterial activities of three cyclic peptides isolated by mRNA display screening. We then determine the X-ray crystal structures of the peptides bound to *E. coli* BamA. We find that each peptide binds to the lumen of the BamA β-barrel at a unique location that differs from the binding sites of previously characterized inhibitors and impairs the biogenesis of several model *E. coli* OMPs both in live cells and in an in vitro BAM-dependent OMP assembly assay. Interestingly, we also find that while the assembly of an OMP mutant that inserts into the OM very slowly is blocked when the peptides are added before it is synthesized, its assembly is accelerated when the peptides are added after it binds to BamA. This result, together with molecular dynamics (MD) simulations that indicate that the peptides promote the closure of the BamA lateral gate, strongly suggests that the peptides function by a previously unreported mode of action and potentially reveals a previously unidentified step in the OMP assembly pathway.

## Results

### Selection of cyclic peptides that bind to BamA through mRNA display screening

To screen for cyclic peptides that bind to the BamA β-barrel, we utilized the PeptiDream Peptide Discovery Platform System (PDPS) (Fig. 1A). The PDPS uses a reconstituted in vitro translation system to incorporate non-canonical amino acids during the synthesis of the ribosomal peptide library. The most critical non-canonical functional group is an *N*-chloroacetyl (ClAc) group introduced at the N-terminus of the nascent peptide by incorporation of a ClAc-bearing amino acid reassigned to the AUG start codon. This functionality enables spontaneous macrocyclization of the peptide scaffold by $S_N2$ substitution of the chloride leaving group with the nucleophilic sulfhydryl group of

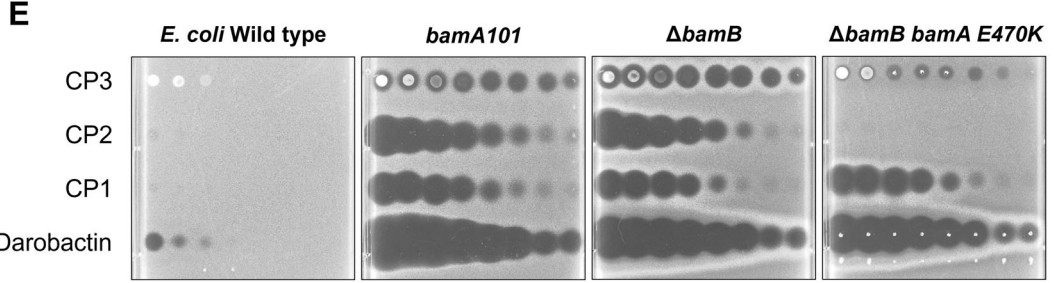

**A**

1. Puromycin linker annealing
2. Translation and conjugation
3. Reverse transcription
4. Panning
5. PCR
6. Transcription

Spontaneous cyclization via ClAc linker

Puromycin

Bead-bound BamA

**B**

CP1: F.L.W.MePhe.Y.H.R.P.V.R.G.MePhe.D.C.G.NH2

CP2: F.S.G.R.W.Tic.W.P.S.R.S.V.G.C.G.NH2

CP3: F.R.R.Y.PeGly.L.D.N.Y.W.V.Tic.Y.C.G.NH2

**C**

(SPR sensorgrams: Response vs Time (s) for CP1, CP2, CP3)

**D**

| Peptide | $K_D$ (nM) | $k_a$ (1/M·s) | $k_d$ (1/s) | MIC (µM) *E. coli* WT | MIC (µM) *E. coli* BamA101 |
|---|---|---|---|---|---|
| CP1 | $6.6 \pm 0.9$ | $3.5 \pm 3.8*10^5$ | $2.3 \pm 2.3*10^{-3}$ | >50 | 12.5 |
| CP2 | $2.0 \pm 1.7$ | $4.3 \pm 2.9*10^5$ | $5.7 \pm 1.9*10^{-4}$ | >50 | 6.3 |
| CP3 | $2.5 \pm 0.2$ | $1.6 \pm 0.3*10^5$ | $3.8 \pm 0.9*10^{-4}$ | >50 | 6.3 |
| Darobactin | $18.1 \pm 3.0$ | $6.3 \pm 2.0*10^5$ | $1.2 \pm 0.5*10^{-2}$ | 4.0 | 0.125 |

**E**

*E. coli* Wild type    *bamA101*    Δ*bamB*    Δ*bamB bamA E470K*

CP3
CP2
CP1
Darobactin

a C-terminal cysteine. In addition, chemically diverse non-canonical amino acids can be incorporated during the elongation of the nascent peptide, increasing the chemical landscape of the macrocyclic peptide library. By using a nucleic acid template library we can generate over $10^{12}$ unique compounds, and by tethering each macrocyclic peptide to its cognate mRNA with a puromycin linker (Fig. S1A) we can both amplify the appropriate cDNAs and identify the sequences of peptides of interest after multiple rounds of in vitro selection[55,56]. All peptides in our library were designed to contain ClAc-L-Phenylalanine (ClAc-F) at the N-terminus, a 12 or 13 amino acid random amino acid region encoded by sequential NNU codons (where N is any of the four RNA nucleosides, and U is uracil), a cysteine, and finally, a GGGGSS linker. Peptides of interest originated from two libraries named "NNU1" and "NNU3" that encode N-methylated and N-alkylated amino acids and

**Fig. 1 | Peptides discovered by mRNA display screening inhibit the growth of BAM-deficient *E. coli* strains. A** Macrocyclic peptides that bind to the BamA β-barrel with high-affinity were identified from an mRNA library of >$10^{12}$ unique sequences using iterative rounds of translation, panning, and amplification. (1) A puromycin-DNA linker was annealed to each mRNA molecule, which serves to (2) tether the nascent macrocyclic peptide to the mRNA that encodes it during in vitro translation. (3) Complementary DNA (cDNA) was reverse transcribed for downstream PCR amplification and for preventing recovery of undesired aptamers during panning. (4) Peptide-mRNA-cDNA complexes were incubated with bead-bound BamA to select for peptides with affinity to BamA. Starting from round 2, a magnetic bead-bound negative selection target and/or free magnetic beads was introduced in a counterscreen to remove undesired peptides prior to incubation with magnetic bead-bound BamA. Peptide-mRNA-cDNA complexes bound to BamA were then recovered and unbound peptides-mRNA-cDNA complexes were washed away. (5) The total cDNA of captured peptide-mRNA-cDNA complexes was quantitated using qPCR, and (6) the bulk recovered cDNA was amplified using PCR for the next round of selection or for sequencing. Next-generation sequencing was performed on the cDNA after each round of noticeably elevated peptide-mRNA-cDNA capture. **B** Sequences of the peptides used in this study. Unnatural amino acids in these peptides include MePhe (N-methyl phenylalanine), Tic (tetrahydro-isoquinoline-3-carboxylic acid), and PeGly (phenethylglycine). **C** Representative SPR sensorgrams of the indicated peptides binding to an *N*-terminally biotinylated *E. coli* BamA β-barrel are shown. Peptides were analyzed in single cycle kinetics mode and injected at five concentrations ranging from 12.3 nM to 1 μM in three-fold increments. Binding kinetics were determined using default Biacore T200 calculations. **D** The $K_D$ determined by SPR is reported as the average of 2, 3, or 4 replicates ± SD. The MIC values of the indicated mRNA display peptide or darobactin against wild-type (WT) and *bamA101 E. coli* strains in solution are shown. **E** Whole-cell activity of peptides and darobactin against *E. coli* on agar plates. The indicated peptide (3 μL) was spotted onto the plate in a two-fold dilution series starting at 2500 μM for CP1-CP3 or 400 μM for darobactin. Source data are provided as a Source Data file.

that have been optimized to translate with high efficiency and fidelity. Libraries were differentiated by the composition of canonical and non-canonical amino acids used for the translation of the random regions (Fig. S1B). The canonical amino acids Ser, Tyr, Leu, Pro, Arg, Asn, His, Val, Asp, and Gly were used for both libraries. The non-canonical amino acids MePhe, MeGly, MeNle, and MeAla were used for the NNU1 library, and PeGly, MeGly, HxGly, and Tic were used for the NNU3 library. For each round of in vitro selection, a biotinylated *E. coli* BamA β-barrel was immobilized on magnetic beads. Enrichment was observed after six rounds of in vitro selection (Fig. S2), and individual clones with a high percentage representation in any round were picked for follow-up single-clone triage. Individual clones were expressed in vitro and assessed for binding to the *E. coli* BamA β-barrel using electro-chemiluminescence (Fig. S3). Multiple macrocyclic peptides that bind to the BamA β-barrel with high affinity were identified and were chemically synthesized. Based on the results of preliminary binding and growth inhibition experiments, three peptides with similar functional profiles (**CP1, CP2**, and **CP3**; see Fig. 1B), were selected for further study.

## Cyclic peptides identified by mRNA display bind specifically to *E. coli* BamA and are antibacterial

We first assessed the ability of the peptides to bind to N-terminally biotinylated *E. coli, Pseudomonas aeruginosa,* and *Acinetobacter baumannii* BamA β-barrels using surface plasmon resonance (SPR). CP1, CP2, and CP3 all bound to the *E. coli* BamA β-barrel with a $K_D$ in the low-nanomolar range (Fig. 1C, D). Using the same method, we found that darobactin has a $K_D$ of ~18 nM. The binding constants of the mono-cyclic peptides did not shift significantly in the presence of darobactin, suggesting that they bind to BamA at different sites (Fig. S4). Interestingly, the peptides did not bind to either *P. aeruginosa* or *A. baumannii* BamA β-barrels (Fig. S5) and therefore lack the broad-spectrum activity of darobactin. This observation also suggests that the peptides bind to BamA by a different mechanism than darobactin.

We next examined whether the peptides affect the growth of *E. coli.* None of the peptides affected the growth of a wild-type *E. coli* strain JCM158 (MC4100 *ara'*) in agar or liquid culture-based assays (Fig. 1D, E). However, the peptides did inhibit the growth of two *E. coli* strains that have compromised BAM activity with a minimum inhibitory concentration (MIC) of ~6.3–12.5 μM: JCM972 (MC4100 *bamA101*), a strain that produces ~5-10-fold less BamA than wild-type strains[57], and AM710 (MC4100 Δ*bamB*[37]), a strain that lacks the BamB subunit of BAM. The activity of the peptides against these mutant strains suggests that they not only bind to BamA, corroborating the SPR binding data, but also impair its function. We also tested the activity of the peptides against AM711, a derivative of AM710 that contains a BamA gain-of-function mutation (E470K) that both bypasses the need for BamD to catalyze OMP assembly and confers resistance to MRL-494, a small

molecule that appears to impair the activity of BamA[16,37]. The observation that the E470K mutation suppressed the antibacterial activity of CP2 and CP3 but not CP1 (Fig. 1E) suggests that CP2 and CP3 interact with BamA differently than CP1. To confirm that the antibacterial activity of CP1-CP3 against JCM972 was due to the binding of the peptides to BamA and not simply to the loss of membrane integrity that results from the *bamA101* and Δ*bamB* mutations, we also measured the activity of the peptides against *E. coli* MB5746, a *bamA+* strain that has a more permeable OM than JCM972 and that is defective in drug efflux[58]. The peptides inhibited the growth of MB5746, but less effectively than the growth of JCM972 (Fig. S6). This observation indicates that the toxicity of the peptides does not correlate solely with the permeability of the OM and strongly suggests that they act by inhibiting BamA function directly. Finally, consistent with the binding results described above, we found that the peptides did not affect the growth of *A. baumannii* or *P. aeruginosa* or the Gram-positive organism *Staphylococcus aureus* (Fig. S6).

## Cyclic peptides CP1, CP2 and CP3 bind to distinct sites in the BamA lumen

Because darobactin does not appear to affect the ability of the cyclic peptides to bind tightly to BamA, we next determined the crystal structure of each peptide bound to the *E. coli* BamA β-barrel to further understand its mode of binding and functional inhibition. For two of the structures, we crystallized the BamA β-barrel bound to darobactin and then soaked in either CP1 or CP2, and for the third structure we co-crystallized the BamA β-barrel with CP3. In all of the structures we observed electron density for the ligands and found that the BamA lateral gate is in a closed conformation, as it is in the published structure of darobactin bound to BamA (Figs. 2A–D and S7, Table S1)[44] and as it is when darobactin binds to BamA in live cells[59]. Comparison of the overall β-barrel structure of peptide-bound and darobactin-bound BamA (PDB 7NRE[44], Fig. 2A) reveals little change, with root-mean-square deviations (RMSDs) between Cα positions of each structure ranging from 0.246–0.456 and virtually no change in the binding mode or conformation of darobactin (Fig. 2A–D, Table S2, Fig. S8). Interestingly, electron density for all three peptides was observed in the BamA β-barrel lumen, distant from the darobactin binding site (Fig. 2B–D and S7). The similarities between the structures presented here (with and without darobactin), the previously reported darobactin-bound structure[44], and several apo-BamA structures in the closed lateral gate state[24,60,61] all indicate that darobactin binding has little influence on the closed-gate BamA tertiary structure. Our data also show that the binding of the cyclic peptides to the BamA β-barrel is independent of darobactin binding.

The crystal structures revealed two distinct mechanisms by which the cyclic peptides could inhibit BamA activity. CP1 and CP3 seal the lateral gate from the inside of the barrel via interactions with residues

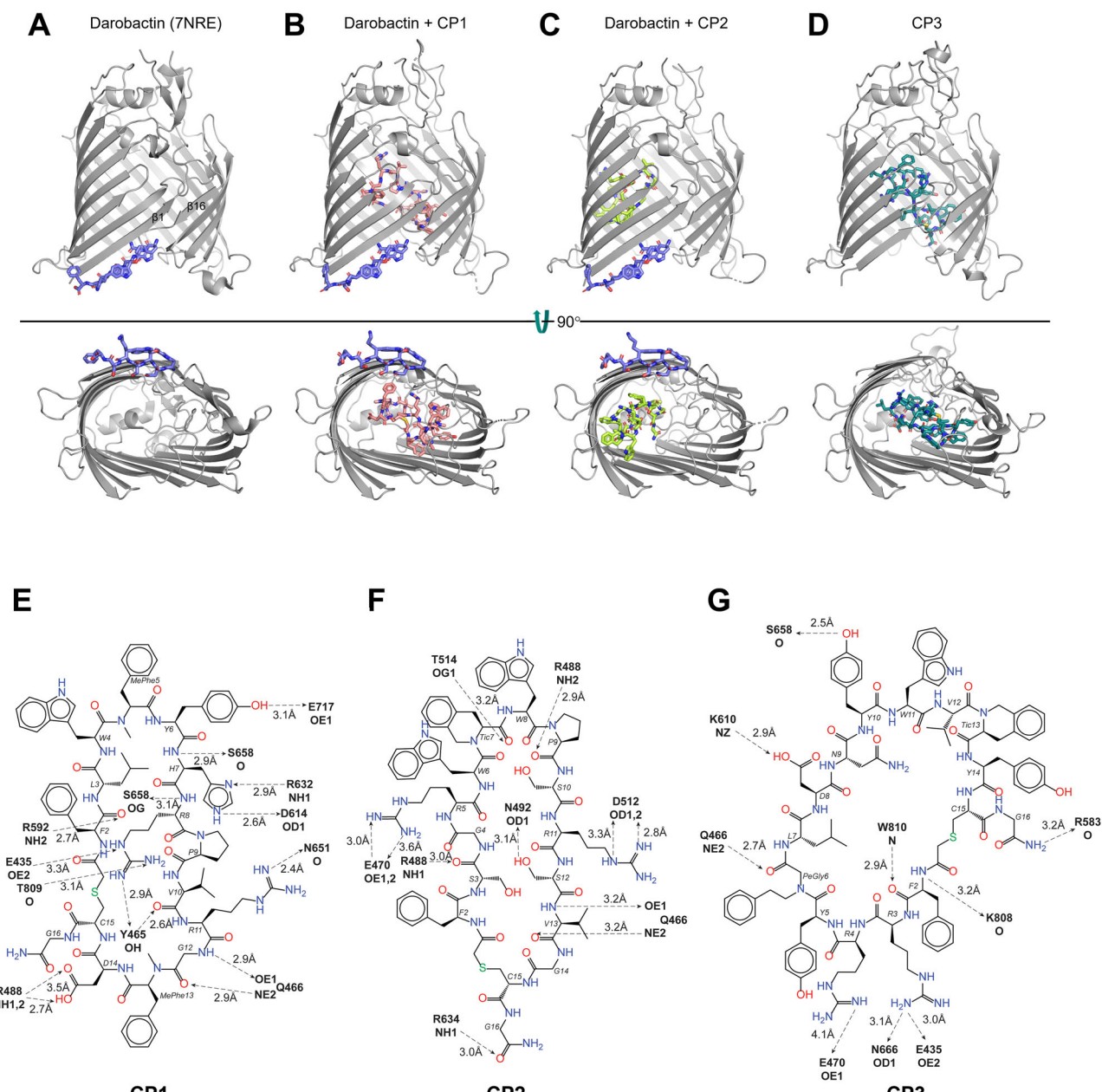

**Fig. 2 | mRNA display peptides bind to the lumen of the BamA β-barrel.** Darobactin (**A**, blue) was previously shown to bind to the lateral gate (strands β1 and β16) of the *E. coli* BamA β-barrel (PDB 7NRE[42]). Our x-ray crystallographic analysis shows that, in contrast, CP1 (**B**, pink), CP2 (**C**, yellow), and CP3 (**D**, teal) bind to the lumen of the BamA β-barrel. In (**A**–**D**) a periplasmic view is shown on the bottom. **E**–**G** The key interactions that mediate the binding of the indicated peptide to the BamA β-barrel are shown.

on both sides of the gate (Figs. 2B, D and S9). Both peptides form primarily polar contacts with BamA, but with different residues. The sidechains of CP1 residues Y6, H7, R8, R11, and D14 form key hydrogen bonds with BamA residues E717, R632/D614, E435/T809/Y465, N651, and R488, respectively (Fig. 2E). The CP1 peptide backbone residues F2 and H7 also make polar contacts with R592 and S658, respectively. In both CP1 and CP3 co-crystal structures, Q466 mediates interactions with the peptide backbone. In contrast, CP3 sidechains R3, D8, and Y10 contact N666/E435, K610, and S658, and thereby form entirely different contacts with BamA than CP1 (Figs. 2D, G and S9). BamA residues K808, W810, Q466, and R583 hydrogen bond with the backbone of CP3 residues F2, PeGly6, and G16, and residue E470 forms a long-distance (>4 Å) electrostatic interaction with CP3 residue R4. The suppression of the antibacterial activity of the peptide by the

BamA(E470K) mutation (Fig. 1E) strongly suggests that the E470-R4 interaction is important. Although the two peptides interact differently with the β-barrel, both compounds trap the C-terminal residue of BamA (W810) in the interior of the barrel (Fig. S10).

Unlike CP1 and CP3, CP2 binds primarily to strands β3 to β7 and does not form interactions with residues at the lateral gate or the BamA C-terminus (Figs. 2C, F, and S9). Three CP2 sidechains, R5, R11, and S12, interact electrostatically or form hydrogen-bonds with BamA residues E470, D512, and N492, respectively (Fig. 2F). Like the interaction of CP3 residue R4 with BamA E470, the interaction of CP2 residue R5 with the same amino acid helps to explain why the E470K mutation suppresses the toxicity of both peptides and supports the idea that their antibacterial activity is a consequence of their interaction with BamA. CP2 backbone residues S3, Tic7, P9, V13, and G16 mediate the remaining

polar interactions with BamA R488, T514, Q466, and R634. Instead of locking the β-barrel in a closed conformation like CP1 and CP3, CP2 might inhibit BamA activity by constraining its flexibility and thereby preventing β1-β8 from folding outward into the lateral-open state. Although the peptides might inhibit BamA function by different mechanisms, it seems likely that their ability to maintain the lateral gate in a closed conformation would prevent the interaction between BamA β1 and the βsignal of client proteins that is required for the initiation of OMP insertion.

We next used molecular dynamics (MD) simulations to compare the stability of two peptides that represent the two modes of inhibition in the lateral-closed state of BamA, CP1 and CP2. We found that over the entire course of three 50 ns simulations RMSD values for both peptides were consistently below 2.5 Å when compared to the crystal structures, demonstrating that they were bound stably to the lumen of the BamA β-barrel (Fig. 3A, B). We also analyzed peptide-BamA residue interaction frequencies to identify the residues that contribute to binding stability (Fig. S11). Residues with a high contact frequency throughout the MD simulations likely form major interactions with the cyclic peptides. Consistent with the notion that the peptides bind stably to their respective locations within BamA in its lateral-closed conformation, each of the high contact frequency residues were within 4 Å of the peptide in the crystal structures (Table S3). Although most of the key interactions identified through MD simulations were also observed in the crystal structures, two additional interactions were observed between CP1 residue R11 and BamA E650 (which can rotate to form an electrostatic interaction) and BamA Y653 (which forms a hydrogen bond) (Fig. S12).

Interestingly, we found that while the BamA side chains that form key interactions with CP1-CP3 are highly conserved in the Enterobacteriaceae, they are much less conserved between *E. coli* and *A. baumannii* and *P. aeruginosa* (Fig. S13). The lack of sequence conservation likely explains why the peptides did not bind to the BamA isoforms produced by these organisms in our SPR experiments (Fig. S5).

## Cyclic peptides identified by mRNA display inhibit BAM activity in vitro and in live cells

To obtain direct evidence that CP1, CP2, and CP3 inhibit BAM activity, we next examined the effect of the peptides on OMP assembly in a previously described in vitro OMP assembly assay[62]. In this assay OMPs are synthesized in a coupled in vitro transcription-translation system in the presence of purified BAM that is reconstituted into lipid vesicles (with the BamA POTRA domains and the BAM lipoproteins facing outward) and the periplasmic chaperone SurA. To detect newly synthesized OMPs, a fluorescent lysine analog is incorporated into the protein during translation. We used three different *E. coli* OMPs that have previously been shown to fold and insert efficiently into BAM proteoliposomes[62,63] as model proteins. The first protein, EspPΔ5′ (previously designated EspP β + 46[64]), is a derivative of a member of the autotransporter family of OMPs (EspP). Autotransporters contain a 12-stranded β-barrel domain and a large extracellular ("passenger") domain that are connected by a linker that traverses the β-barrel lumen. Although the ~35 kDa EspPΔ5′ protein lacks almost the entire passenger domain, this truncation has no effect on β-barrel assembly[65]. EspPΔ5′ assembly can be assessed by monitoring an autocatalytic intra-barrel cleavage reaction that separates the two domains only after the passenger domain is translocated across the OM and the assembly of the β-barrel domain is complete[27,66]. Like most OMP β-barrel proteins, the cleaved EspP β-barrel is also "heat modifiable" if it is properly folded; that is, in the absence of heat the protein is resistant to SDS denaturation and consequently migrates on SDS-PAGE more rapidly than its predicted molecular weight. The other two proteins we used are OmpT, which folds into an empty 10-stranded ~32 kDa β-barrel, and OmpC, which

forms a trimeric porin composed of three identical 16-stranded ~38 kDa β-barrels. We assessed the folding of both proteins by monitoring heat modifiability; in the case of OmpC the three subunits remain together on SDS-PAGE in the absence of heat and migrate well below the predicted ~115 kDa molecular weight.

Despite the observation that the three cyclic peptides only affect the growth of *E. coli* strains in which BAM is compromised (Fig. 1D, E), we found that all three peptides dramatically inhibited OMP assembly in vitro. To maintain consistency with previous studies[62,63], 0.5 μM BAM reconstituted into 1-palmitoyl-2-oleoyl-glycero-3-phosphocholine (BAM-POPC) was added to each reaction. To match or exceed the BAM concentration, the peptides were added at a concentration of 100x−800x $K_D$ (0.25−6 μM). Consistent with previous results[62,63], in the absence of peptide ~50% of the EspPΔ5′, ~40% of the OmpT, and ~30% of the OmpC were assembled (Fig. 4A, C and E, lanes 1−2). Remarkably, CP1 partially inhibited OMP assembly at a concentration of 0.75 μM and almost completely blocked OMP assembly at a concentration of 1.5 μM (Fig. 4A, B). Likewise, CP2 and CP3 strongly inhibited OMP assembly at concentrations of 1.8 μM and 1.0 μM, respectively (Fig. 4C−F). All of the compounds inhibited the insertion of OmpC monomers into BAM-POPC vesicles (Fig. S14). Dynamic light scattering (DLS) experiments showed that the peptides did not alter the size of the proteoliposomes and thereby rule out the possibility that the peptides inhibited BAM function by disrupting the integrity of the vesicles (Fig. S15). By comparison, in the same assay darobactin strongly inhibited the assembly of EspPΔ5′ and OmpC at concentrations of 5 μM and 2 μM, respectively, and only blocked the assembly of one protein (OmpA, an 8-stranded empty barrel) at a concentration of 1 μM (Fig. S16A). Like our cyclic peptides, darobactin did not disrupt the structure of the vesicles (Fig. S16B). In contrast, the putative BamA inhibitor Polyphor 7 did not inhibit BAM activity even at a concentration of 400 μM ( ~ 2000 x $K_D$) presumably because it binds to the external loops of BamA that are located inside the proteoliposomes (Fig. S17).

Consistent with our in vitro data, we found that CP1, CP2 and CP3 also inhibit BAM activity in live cells, but less effectively. To assess the effect of the cyclic peptides on OMP assembly, we first transformed HDB164, a strain that contains the *bamA101* allele with a plasmid that encodes *espPΔ5* under the control of a rhamnose-inducible promoter. After the expression of *espPΔ5* was induced for 5 min, cells were subjected to pulse-chase labeling to monitor the fate of newly synthesized EspPΔ5 and two chromosomally encoded OMPs, OmpC and OmpA. Radiolabeled cells were harvested at various timepoints, and the three proteins were immunoprecipitated and resolved by SDS-PAGE. Five min prior to the start of the chase (i.e., 4.5 min prior to pulse labeling), cells were treated with one of the cyclic peptides dissolved in DMSO at an optimal concentration (2x MIC = 12.5 μM or 25 μM) or mock-treated with an equal amount of DMSO. Based on the accumulation of the cleaved EspPΔ5 β-barrel in the treated and the mock-treated (control) samples, all of the cyclic peptides inhibited the assembly of this protein by ~82−90% (Fig. 5A, B, top gels and left graphs). The finding that the total signal generated by preproEspPΔ5 (the form of the protein that contains the signal peptide, the passenger domain and the β-barrel), proEspPΔ5 (the form of the protein that contains the passenger domain and β-barrel), and the cleaved β-barrel was much lower at the 10 min timepoint than at the start of the chase indicates that in the presence of the cyclic peptides most of the protein was trapped in the periplasm and degraded. Likewise, the observation that almost all of the OmpC was degraded at the 10 min timepoint in cells treated with CP1, CP2 and CP3 but was stable in the control cells shows that no more than a small percentage of the protein was assembled in the presence of the cyclic peptides (Fig. 5A, B, middle gels and middle graphs). Based on its stability, however, the assembly of OmpA was only partially inhibited by the peptides (Fig. 5A, B, bottom gels and bottom graphs). Finally, to confirm that the cyclic peptides target BAM and do not inhibit OMP assembly by an indirect mechanism, we examined

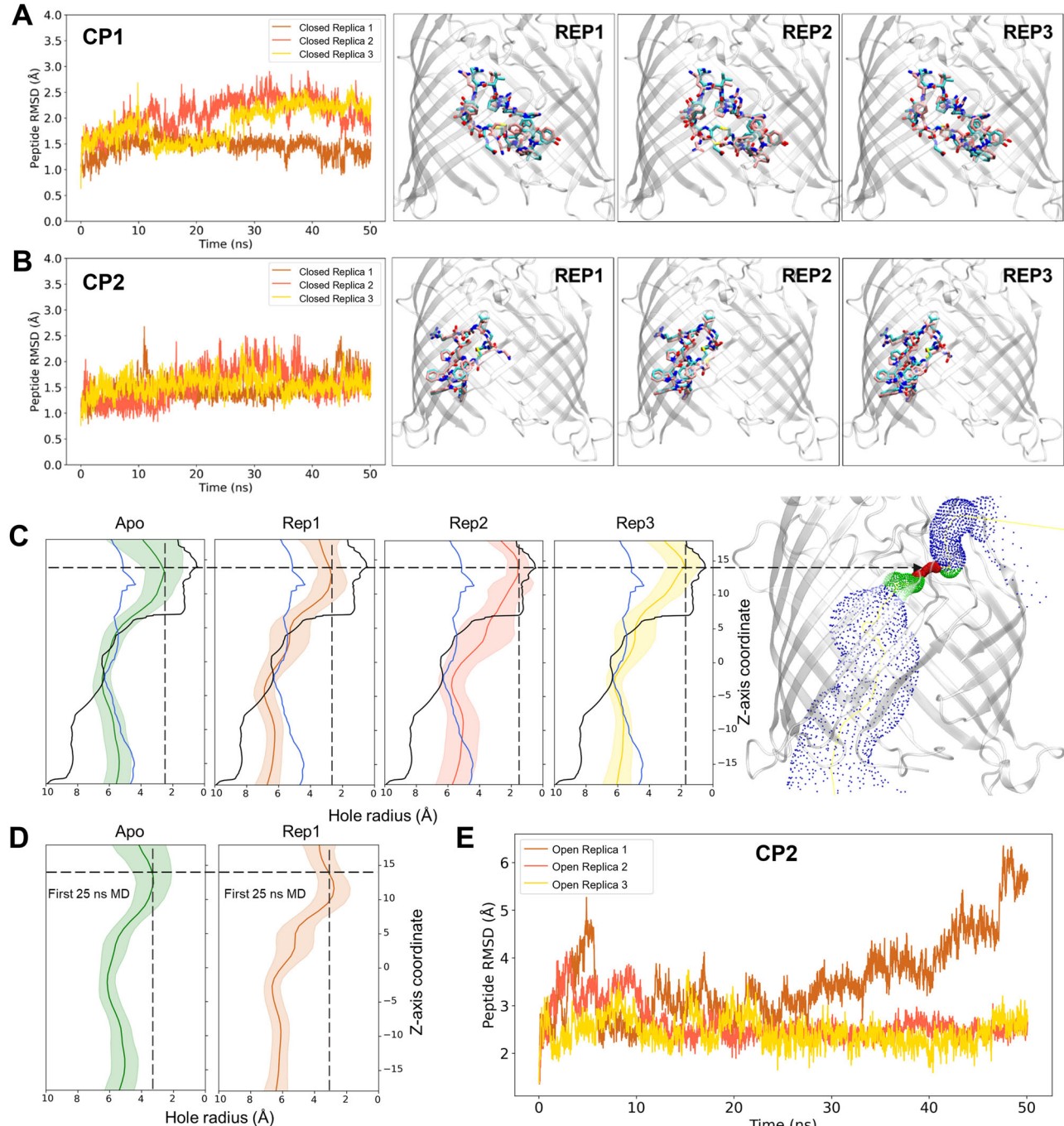

**Fig. 3 | MD simulations of peptide-BamA interactions.** RMSD of lumen binders over the course of the simulation demonstrate the conformational stability of **A** CP1 and **B** CP2 in the closed state. Snapshots of CP1 and CP2 in three replicates are displayed on the right. The initial position of the peptides and their position after 50 ns are shown in cyan and pink, respectively. **C** The hole radius profile for the Apo BamA closed (PDB 9CS0) and open (PDB 5LJO[22]) structures are represented by the black and blue lines, respectively. The computation of the average hole radius from the 50 ns of the MD simulations for Apo BamA starting in the open state are shown in green, and the three replicates of BamA + CP2 are shown in brown, red, and yellow. Crosshairs indicate the hole radius of each simulation at a Z-coordinate of 14 Å. **D** The computation of the average hole radius from the first 25 ns of the MD simulations for Apo BamA (green) and replicate 1 of BamA + CP2 (brown) are shown. **E** RMSD values of the CP2 peptide bound to BamA in an open state over the course of the three 50 ns simulations are shown. The data in **C** and **D** are presented as mean values and error bars represent the standard deviation.

their effect on the assembly of EspPΔ5 in a wild-type *bamA*+ strain (AD202). As we predicted, the compounds inhibited EspPΔ5 assembly by only ~50–60% (Fig. S18).

The observation that the cyclic peptides inhibited BAM activity at a much lower concentration in vitro than in live cells strongly suggests that the peptides can access the lumen of the BamA β-barrel from the periplasmic side of the OM more readily

than from the extracellular side. Based on our results, it seems likely that the peptides, which have a molecular weight of >2000 Da, cannot cross the OM efficiently through the pores formed by porins or by an alternative pathway. Indeed the intermediate sensitivity of strain MB5746 to the peptides (Fig. S6) might be due to the permeability of its OM. Although OMP substrates enter the BamA β-barrel from the periplasmic side, there has not been any

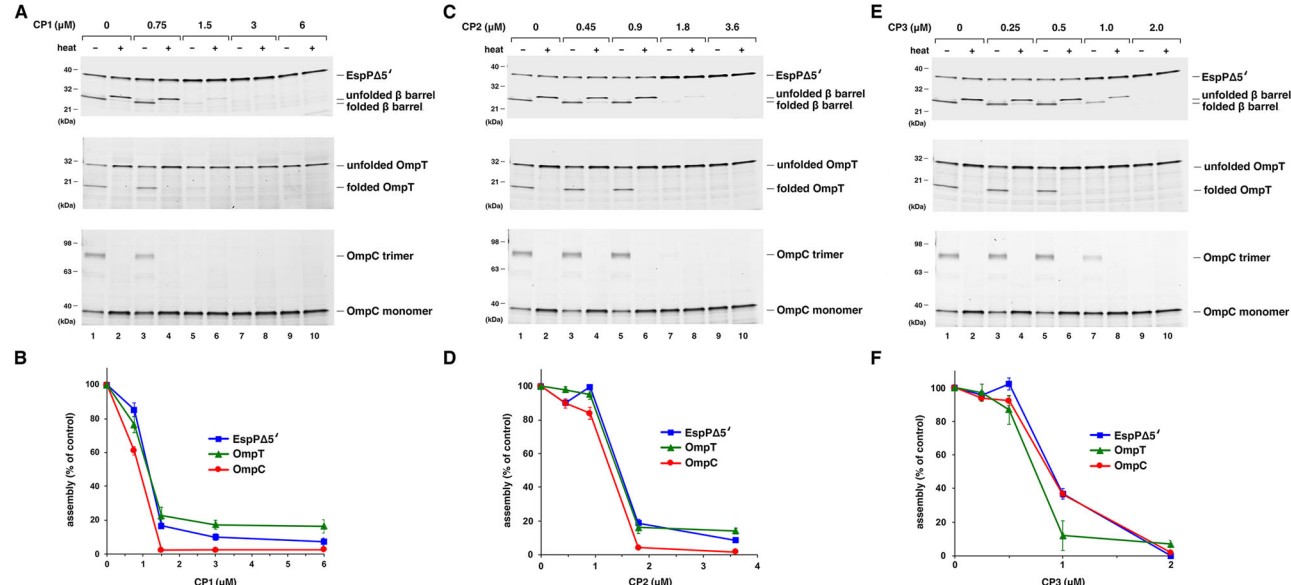

**Fig. 4 | Cyclic peptides CP1, CP2 and CP3 effectively inhibit the assembly of model OMPs in vitro.** PURExpress coupled transcription/translation reactions supplemented with BODIPY-FL-ε-Lys-tRNA$^{lys}$, 2 μM SurA and 0.5 μM BAM-POPC (and 0.5 mM Rc LPS for reactions in which OmpC was synthesized) were incubated at 30° C for 5 min with the indicated concentrations of cyclic peptide CP1 (**A**, **B**), CP2 (**C**, **D**) or CP3 (**E**, **F**). Samples were then programmed with a plasmid that encodes EspPΔ5′, OmpT, or OmpC and incubated at 30° C for 30 min. Subsequently half of each reaction was heated to 95° C while the other half was left unheated, and proteins were resolved by SDS-PAGE. Newly synthesized OMPs were detected based on the incorporation of fluorescently tagged lysine residues. Representative gels are shown in **A**, **C** and **E**, and the concentration-dependent effect of the peptides on the assembly of each OMP in three biochemically independent experiments is shown in **B**, **D**, and **E**. The percent of the protein that was assembled in the absence of peptide is defined as 100%. The heated samples were used to quantitate EspPΔ5′ assembly while the unheated samples were used to quantitate the assembly of the other OMPs. The data are presented as mean values and error bars represent the SEM. Source data are provided as a Source Data file.

---

indication that lumen binders (or even darobactins that bind close to the periplasmic side of BamA) might have a preferred mode of entry.

### CP1, CP2 and CP3 block the binding of substrates to BamA but accelerate a later stage of assembly

To gain insight into the mode(s) of action of the cyclic peptides, we next examined their effect on the biogenesis of an EspP mutant that assembles extremely slowly in live cells. Whereas >80% of wild-type EspP is assembled within 2 min after it is synthesized, the G1123R mutant does not reach the same level of assembly until 20 min, and the mutant continues to assemble until at least 30 min[67]. Like wild-type EspP, the mutant protein binds to BAM and the β-signal of the protein binds to BamA β1 rapidly. Interestingly, the mutant remains relatively stable during the entire assembly period. Consistent with these observations, darobactin completely blocks the assembly of EspP(G1123R) and promotes its degradation when the BamA inhibitor is added to cultures before the mutant protein is synthesized, but has no effect on assembly when added to cultures after the protein binds to BAM[41]. These results and the observation that the exposure of the passenger domain on the cell surface parallels the completion of assembly[67] strongly suggest that while the β-signal of the mutant effectively binds to BamA β1 to initiate assembly, the lipid facing arginine at position 1123 inhibits the full insertion of the β-barrel into the OM. Indeed when we treated permeabilized cells with proteinase K (PK) we observed ~26–28 kDa and ~12–14 kDa C-terminal fragments that provide direct evidence that at least part of the EspP (G1123R) β-barrel remains exposed for up to 30 min after the β-signal binds to BamA (Fig. S19). We studied the effect of the cyclic peptides on EspP(G1123R) assembly in a *bamA+* strain (AD202) because the mutant protein reduces the availability of BAM by remaining bound to BamA for a prolonged period and is consequently highly toxic in *bamA101* strains.

Consistent with our analysis of the effect of CP1, CP2, and CP3 on the assembly of several wild-type OMPs in live cells, the peptides strongly inhibited the assembly of the EspP(G1123R) mutant in pulse-chase experiments when added at 2x MIC 5 min prior to the start of the chase (Fig. 6A). Most of mutant protein was degraded in the periplasm, and only a small fraction underwent proteolytic processing (Fig. 6B, compare lanes 1–5 to 6–10; partly because of the instability of the protein its assembly cannot be accurately quantitated). Surprisingly, however, we found that unlike darobactin, the cyclic peptides significantly accelerated the assembly of EspP(G1123R) when added 2 min after the start of the chase (Fig. 6A) and as a consequence slightly increased the stability of the mutant protein (Fig. 6B, compare lanes 1–5 to 11–15 and the two curves in the graphs on the right; Fig. S20). Based on the observation that both darobactin (a β-signal mimetic) and either CP1 or CP2 can bind to BamA simultaneously (Fig. 2B, C), it is likely that the peptides exert this effect by binding to the BamA lumen while the β-signal of the EspP mutant is bound to BamA β1. This scenario was supported by the results of site-specific crosslinking experiments that confirmed that the EspP(G1123R) mutant associates with BAM before the 2 min timepoint and that the cyclic peptides do not promote its dissociation (Fig. S21). Even more remarkably, the effect of the cyclic peptides phenocopied the effect of adding 2.5 mM EDTA to cells that produced the EspP(G1123R) mutant (but not wild-type EspP) at either the same timepoint or at 1 min or 5 min prior to the start of the chase (Figs. 6C and S22). As expected, the EDTA effectively reduced the fluidity of the OM even in the absence of Tris buffer by releasing LPS into the culture medium, but the cyclic peptides did not release any LPS (Fig. 6D). A similar acceleration of EspP(G1123R) assembly was observed when 0.8 M sorbitol (which creates a hypertonic condition that relaxes OM tension) was added at the 2 min timepoint (Fig. 6E). Interestingly, consistent with the observation that membrane tension plays a role in late steps of OMP assembly[25], sorbitol appeared to slightly delay the assembly of wild-type EspP (Fig. S23).

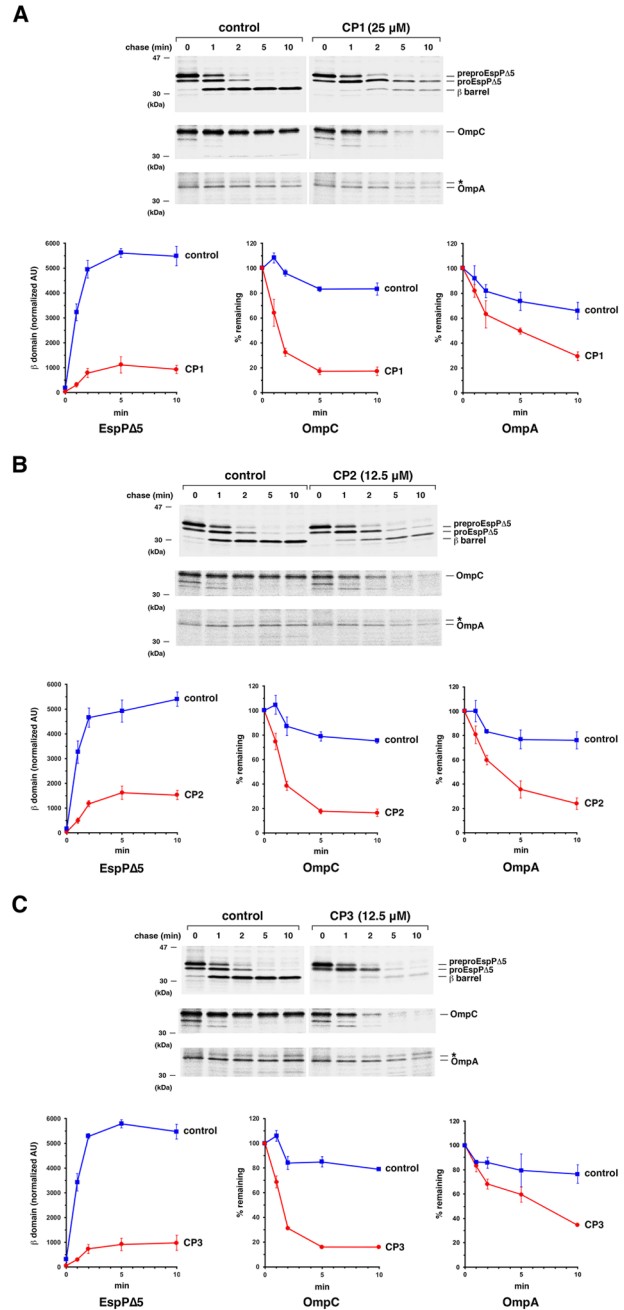

**Fig. 5 | Cyclic peptides CP1, CP2, and CP3 inhibit the assembly of model OMPs in a *bamA101* strain.** HDB164 (MC4100 *ara^r bamA101 ompT::spc*) transformed with plasmid pJH110 (P*_{trc}*-*espP*Δ5) were grown in M9 to log phase, the expression of *espP*Δ5 was induced by the addition of 10 μM IPTG, and cells were subjected to pulse-chase labeling. Five min prior to the start of the chase the indicated amount of CP1 (**A**), CP2 (**B**) or CP3 (**C**) dissolved in DMSO, which is equivalent to 2x MIC, or DMSO alone (control) was added to the cultures. Immunoprecipitations were performed using antisera directed against EspP, OmpA and OmpC peptides and proteins were resolved by SDS-PAGE. Representative gels are shown in each panel, and the mean values of the results obtained in three independent pulse-chase experiments are presented in the graphs below the gels. Error bars show the SEM. The assembly of EspPΔ5 was quantitated by normalizing the βbarrel domain that had accumulated at each time point to the total signal observed at the 0 min time point (in arbitrary units, AU). The assembly of the other OMPs was quantitated by determining the percent of the radiolabeled protein that remained at each time point. The asterisk denotes an unknown background band. Source data are provided as a Source Data file.

These results not only strongly support the notion that the three cyclic peptides inhibit BAM activity by a previously unreported mode of action, but suggest more specifically that by binding to BamA the compounds mimic conditions that lower the kinetic barrier for OMP insertion and thereby promote the post-initiation step at which the assembly of EspP(G1123R) is rate-limited.

### The lumen-binding cyclic peptides potentially promote the transition of BamA from an open to a pre-closed state

To shed light on the mechanism by which the cyclic peptides might accelerate the assembly of the EspP(G1123R) mutant, we used MD simulations to observe the trajectories of CP1 and CP2 after docking them into the BamA β-barrel in its lateral open conformation (PDB 5LJO). To study the dynamics of BamA in the presence and absence of peptides, we utilized the BamA hole radius, which represents the radius of the largest sphere that could fit into the β-barrel at a given Z-coordinate, as a metric for openness. For example, the smallest hole radius in the closed state crystal structure is <1 Å at a Z coordinate of -14 Å (Fig. 3C, black line). In contrast, the hole radius at the same Z coordinate expands to ~5 Å in the open state structure (Fig. 3C, blue line). To characterize the dynamic changes of BamA during MD simulations, we extracted 2500 frames from the 50 ns simulations and calculated the hole radius of each frame. As a control, we first measured the hole radius of apo BamA over a 50 ns MD simulation starting from the open state structure and observed an average radius of ~2.3 Å at a Z coordinate of ~14 Å. This value is much lower than the 5 Å observed in the open state crystal structure (Fig. 3C, Apo). We then observed the radius for the BamA open state simulation after docking in CP2. In two of three replicates, the peptide remained stable during the entire 50 ns simulation and the average hole radius was smaller than that of apo BamA (i.e., <2 Å at Z = 14 Å; Fig. 3C, replicates 2 and 3). The decrease in hole radius suggests that the peptide facilitates faster lateral gate closing. In one replicate, however, we observed a slightly larger hole radius than in apo BamA (Fig. 3C, replicate 1), but an increase in peptide RMSD during the last 25 ns of the simulation indicated that the CP2 docking pose was relatively unstable during that time window (Fig. 3E). To determine whether the peptide expedited closing when it was stably bound, we compared the hole radius profile within the first 25 ns of replicate 1. As we observed in the other two replicates, the minimum average hole radius was smaller in the presence of CP2 (<3 Å at Z = 14 Å) than in its absence (>3 Å at Z = 14 Å) within the shorter timeframe (Fig. 3D). The results of replicate 1 therefore are consistent with the possibility that the stable binding of CP2 accelerates the conformational change of BamA from an open to a pre-closed (or partially closed) state.

Curiously, in three replicate MD simulations in which CP1 was docked into open state BamA, the average hole radii ranged from 3.5–4 Å (at Z = 14 Å), which was greater than the average hole radius observed in the apo BamA simulation (~2.3 Å) (Fig. S24A). An increasing peptide RMSD over the full course of the simulation, however, showed that the CP1 docking pose was unstable (Fig. S24B). The instability is likely attributable to an inaccurate initial docking position of the peptide within BamA in its open state. Indeed because the crystal structure shows CP1 bound across the lateral gate of BamA in its closed state (Fig. 2B), it is difficult to dock the peptide into the open state structure. Although the results suggest that CP1 is unable to facilitate the closure of BamA directly from its open state, it is possible that CP1 binds solely to a more closed state of BamA, locking the β-barrel into an intermediate conformation. Taken together with our experimental results, the MD simulations suggest that the peptides that bind to the lumen of the BamA β-barrel accelerate the assembly of EspP(G1123) at a post-initiation stage by either expediting the transition from an open to a pre-closed state or temporarily trapping the β-barrel in a pre-closed state.

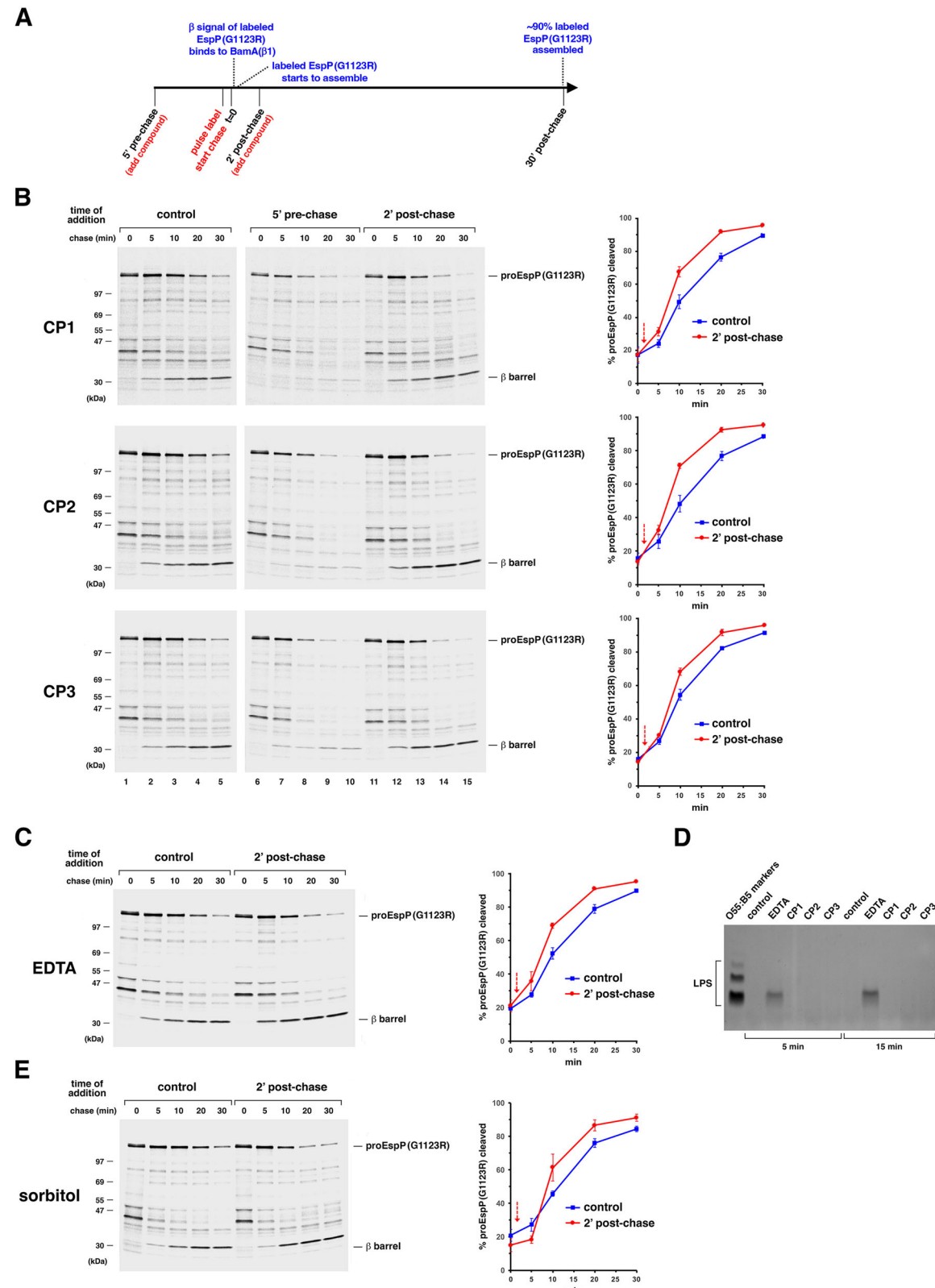

## Discussion

In this report, we describe the use of mRNA display screening to identify cyclic peptides that both bind to the *E. coli* BamA β-barrel and exhibit antibacterial activity by inhibiting BAM function. Given that mRNA display provides a method to generate a library of >10^12 unique cyclic peptides and that cyclic peptides that bind to multipass transmembrane targets with a nanomolar affinity have been described in

multiple studies[53,68–72], it is not surprising that we identified peptides that bind to the BamA β-barrel with high affinity. It was not immediately apparent, however, if the high affinity binders would also inhibit BAM activity. Consistent with this notion, the cyclic peptides we studied were antibacterial when added to *E. coli* strains that have BAM deficiencies, and they inhibited the assembly of several model OMPs both in live cells and in an in vitro assay in which purified BAM

**Fig. 6 | The cyclic peptides, EDTA and sorbitol accelerate the assembly of EspP(G1123R) if added after the mutant binds to BAM. A** A schematic outlining the procedure used in **B**, **C**, and **E** (red) and the biogenesis of the population of EspP(G1123R) molecules synthesized during the pulse labeling period (blue). AD202 (MC4100 *ompT::kan*) transformed with pJH224 [P$_{trc}$-*espP*(G1123R)] were grown to log phase in M9, the expression of *espP*(G1123R) was induced by adding 10 μM IPTG, and cells were subjected to pulse-chase labeling. **B** The indicated peptide was added at 2x MIC 5 min before the start of the chase or 2 min after the start of the chase or only DMSO was added (control). Immunoprecipitations were performed using an anti-EspP C-terminal peptide antiserum and proteins were resolved by SDS-PAGE. Representative experiments are shown (left), and the completion of EspP(G1123R) assembly was quantitated by calculating the percent of radiolabeled proEspP(G1123R) that underwent autocatalytic cleavage in untreated cells (control) or cells treated with the indicated peptide 2 min after the start of the chase (red arrows) in three biologically independent experiments (right). The

assembly of EspP(G1123R) in cells that were treated 5 min pre-chase cannot be accurately quantitated due to the instability of the protein. The data are presented as mean values and error bars represent the SEM. In **C** 2.5 mM EDTA was added 2 min post-chase, and in **E** cells were harvested and resuspended in M9 (control) or M9 containing 0.8 M sorbitol 2 min post-chase. A representative experiment is shown (left), and the assembly of EspP(G1123R) in cells treated with EDTA or sorbitol or untreated (control) in three biologically independent experiments was quantitated as in **B** (right). **D** AD202 transformed with pJH224 were grown and the expression of *espP*(G1123R) was induced as described above and harvested 5 or 15 min after they were treated with 2.5 mM EDTA or the indicated peptide at 2x MIC or untreated (control). The culture medium was subjected to SDS-PAGE and LPS that was released from the cells was detected by staining. *E. coli* O55:B5 LPS (1 μg) was used as a positive control and molecular weight marker. This experiment was performed twice with similar results. Source data are provided as a Source Data file.

catalyzes the assembly of OMPs into proteoliposomes. The antibacterial activity of CP2 and CP3 was suppressed by a BamA mutation (E470K) that alters a residue with which the peptides form bonds and that was previously shown to suppress the activity of a different type of compound[37]. Furthermore, all three peptides inhibited the assembly of model OMPs less efficiently when added to a wild-type strain than to a BamA-deficient strain. Interestingly, we found that although the cyclic peptides all bind stably to distinct sites in the lumen of the BamA β-barrel, unlike darobactin, they inhibit BAM activity at a significantly lower concentration when they have direct access to the periplasmic side of the BamA β-barrel. This result together with the high MIC values for wild-type *E. coli* strongly suggests that the cyclic peptides either enter the BamA lumen from the periplasmic side but cross the OM poorly because they are too large and hydrophobic to be transported effectively through porins, or that they can enter the BamA lumen from the extracellular side, but relatively inefficiently. In any case, the data strongly suggest that even though BamA must open laterally to catalyze OMP assembly, it does not form a simple channel that provides bidirectional access to small cyclic peptides.

We obtained considerable evidence that CP1, CP2 and CP3, which are the only closed-state BamA lumen binders reported to date, all inhibit BAM activity by a previously undescribed mechanism. None of the peptides bind to the same site as darobactin, which binds predominantly to BamA β1 and thereby functions as a competitive inhibitor of β-signal binding, Polyphor7, which binds to external loops of BamA, or two recently-reported peptide inhibitors, one that binds to the exterior of BamA in the closed state and the other that binds to the BamA lumen in the open state[54]. Indeed, both darobactin and at least two of the peptides (CP1 and CP2) can bind to BamA simultaneously. Although it is difficult to imagine that the peptides directly block β-signal binding, like darobactin, they all maintain the BamA lateral gate in a closed conformation that presumably prevents incoming OMPs from forming interactions with BamA that are necessary for assembly. Curiously, CP1 and CP3 lock the gate shut by binding to residues on both sides of the BamA β-barrel seam while CP2 appears to constrain the flexibility of the β-barrel that is required for the formation of a lateral-open conformation. The observation that all of the peptides destabilize model OMPs in live cells if added prior to their synthesis (presumably by blocking the early steps of membrane insertion) strongly suggests that a compound that prevents the opening of the lateral gate will produce similar effects regardless of the mechanism by which it functions.

The observation that the cyclic peptides accelerate the assembly of the EspP(G1123R) mutant when they are added to cells after the mutant protein interacts with BAM provides the strongest evidence that they inhibit BAM activity by a previously unreported mode of action. It is clear that the peptides function very differently than Polyphor7 and MRL-494, which inhibit the assembly of EspP(G1123R) at this stage, and darobactin, which cannot bind to BamA once the

EspP(G1123R) β-signal binds to BamA β1 and has no effect on assembly[41]. Because previously reported results[41,67] and the results shown in Fig. S19 provide strong evidence that the rate limiting step in the assembly of the mutant protein is the complete insertion of its β-barrel into the OM (or the insertion of its β-barrel in the correct configuration), the cyclic peptides presumably enhance the integration process. Based on our findings, it is likely that the peptides function by promoting a conformational state that disfavors the binding of OMP β-signals to BamA β1, but that favors the insertion of β-barrels into the OM after their β-signals are bound to BamA β1. Our observation that the peptides phenocopy the effect of adding compounds that alter the fluidity or rigidity of the OM (even though, unlike EDTA, they do not release LPS into the culture medium and therefore do not appear to change the lipid composition of the OM) strongly suggests that in this conformation BamA perturbs the local membrane environment. Given that the OM is unusually rigid and that BamA can locally distort the OM to reduce the energy requirement for OMP insertion[25,73], BamA might facilitate effective integration of OMPs by transiently thinning the OM or reducing membrane tension in the Z axis. The finding that EDTA does not accelerate the assembly of wild-type EspP strongly suggests that insertion is typically a fast step in assembly. Indeed this notion is consistent with the proposal that OMPs rarely contain lipid facing charged residues positioned near the middle of the OM because they have evolved to assemble rapidly[67].

Taking our observations and previous results into account, we propose that the BamA β-barrel alternates between multiple states during an assembly cycle (Fig. 7). First, BamA opens laterally (or exists in a laterally open ground state) so that the β-signal of an incoming OMP can bind to β1. Perhaps counterintuitively, in the second stage, the BamA β-barrel closes partially and thereby transiently alters the local OM environment to promote the membrane insertion of client OMPs. This assembly step has not been observed previously, and it was detected only through the discovery of peptides that promote the formation or persistence of a conformation in which BamA accelerates the membrane insertion of a slow assembly mutant. The adoption of this conformational state might not significantly affect the assembly of most OMPs because they enter the OM rapidly as a partially curved β-sheet that is antagonistic to the plane of the OM[25], and the downward pressure imposed by lateral membrane tension is sufficient to promote the subsequent formation of a hybrid barrel and barrelization. In contrast, a surface localized arginine might prevent the complete insertion of a β-sheet (or destabilize it even if it is located inside the lipid bilayer) and require a decrease in pressure through increased local fluidity or reduced local tension for assembly to progress. In the third stage the BamA β-barrel opens to form a hybrid barrel with the client OMP and, as mentioned above, OM tension helps β-barrels undergo a conformational transition to a barrelized structure. Indeed our observation that sorbitol slightly delays the assembly of wild-type EspP is consistent with this notion. In the last step the BamA β-barrel

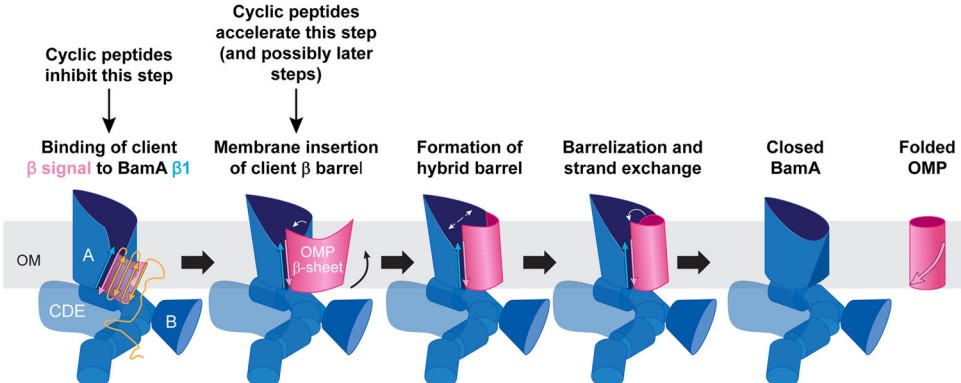

**Fig. 7 | Model for BAM-mediated OMP assembly and the mode of action of the cyclic peptides.** Previous studies have provided strong evidence that the BamA β-barrel is in a laterally open conformation at the start of the assembly cycle (or adopts a laterally open conformation). The lateral opening causes local disorder in the lipid bilayer that likely facilitates OMP assembly. Based on our results, we propose that by binding inside the lumen of the BamA β-barrel, the cyclic peptides CP1, CP2, and CP3 keep the β-barrel in a laterally closed conformation that prevents the binding of client β-signals to BamA β1. By maintaining this conformation, the peptides—and reagents that increase the fluidity or reduce the tension of the OM— accelerate a previously undetected step in the OMP assembly cycle in which the BamA β-barrel partially closes to promote the membrane insertion of OMPs whose β-signal is already tightly bound to BamA β1. Because the membrane insertion of OMPs is generally very fast, we were only able to detect this step through the use of an OMP mutant whose β-barrel domain is fully inserted into the OM (or inserted in a favorable conformation) very slowly. Following membrane insertion, the BamA β-barrel opens more widely to form a hybrid β-barrel with the client protein which then undergoes barrelization. Finally, both the BamA β-barrel and the client β-barrel close through a strand exchange reaction and the folded OMP is released into the lipid bilayer. It should be noted that the membrane insertion of OMPs and the formation of a hybrid barrel (and possibly barrelization/strand exchange) might occur in a concerted reaction, in which case the cyclic peptide might also accelerate later stages of the assembly reaction.

closes to promote closure and release of the client β-barrel. It should be noted that the exact conformation of the BamA β-barrel in the different open and closed states and the degree to which the BamA POTRA domains and BAM lipoproteins contribute to the transitions is unclear. Furthermore, an important caveat to our model is that the membrane insertion of OMPs might not be a distinct step, but an event that is coupled to downstream steps. It is certainly possible that membrane insertion, the formation of a hybrid barrel, and the barrelization/strand exchange steps all occur in one concerted reaction. Indeed a transition of the BamA β-barrel to the lateral closed state after the β-signal of a client protein binds to BamA β1 might drive most of the assembly process.

In any case, the finding that none of the cyclic peptides prevent OMP assembly by blocking the BamA β-barrel lumen is consistent with evidence that BamA does not function as a conventional channel, but rather catalyzes OMP assembly by opening laterally and forming end-joining interactions with client proteins. In contrast, it is surprising that none of the peptides inhibit the translocation of the EspP passenger domain even though it traverses the BamA-EspP hybrid barrel near BamA β1-β3[74]. Given our results, it will be interesting to determine if the EspP linker and other internal segments of OMP β-barrels (e.g., the plug domains of TonB-dependent transporters) that might occupy the hybrid barrel during OMP assembly displace the cyclic peptides.

Our work demonstrates that mRNA display is a promising method to identify new antibiotics that are cost-effective to produce and that complement approaches to identify natural products or synthetic compounds that inhibit the activity of essential bacterial factors by different modes of action. Although CP1, CP2 and CP3 only inhibit the activity of BamA in live cells at relatively high concentrations, they could in principle be linked to bridge peptides to increase their ability to traverse the OM and, based on our in vitro data, thereby dramatically increase their potency. Furthermore, our observation that cyclic peptides can be discovered that selectively target only one or a small subset of Gram-negative organisms through BamA inhibition is highly significant given that narrow-spectrum antibiotics are less likely to harm commensal organisms than broad spectrum antibiotics and would therefore reduce the risk of collateral *Clostridoides difficile* infection and disorders that result from non-specific perturbation of the microbiome. Finally, the recent observation that macrocycles that inhibit BamA activity by different mechanisms than CP1, CP2, and CP3 can be isolated by mRNA display under different target presentation and panning conditions[54] underscores the power of this approach to identify multiple inhibitors of challenging drug targets.

## Methods

### Strains, growth media, and antisera

The *E. coli* K-12 strains used in this study are listed in Table S4. Cells were grown in Luria-Bertani (LB) medium or Terrific Broth (TB) containing the indicated antibiotics for protein expression. For pulse-chase experiments, cells were grown in M9 medium (M9) containing 0.2% glycerol and 40 μg/ml L-amino acids except methionine and cysteine and supplemented with the indicated antibiotics. Rabbit polyclonal antisera raised against C-terminal peptides of EspP and OmpC and OmpA loop 4 have previously been described[75–77].

### Cloning, expression, and purification of the BamA β-barrel for mRNA display screening and SPR

For SPR, *E. coli* K-12 BamA (*Ec*BamA; residues 421–810 from strain MG1655), *A. baumannii* BamA (*Ab*BamA; residues 425–841 from strain ATCC19606) and *P. aeruginosa* BamA (*Pa*BamA; residues 420–797 from strain PAO1) were synthesized and cloned into pET23b (Millipore Sigma) with a N-terminal Avitag for biotin conjugation. For mRNA display screening, an additional *E. coli* BamA (421–810) construct (*Ec*BamA') was designed with an N-terminal His tag, a TEV protease cleavage site, and three point mutations (C690S/C700S/S752C) to allow for site specific biotinylation of residue 752.

The following purification scheme was used for all proteins. *E. coli* BL21(DE3) One Shot competent cells (ThermoFisher catalog number C600003) were transformed with one of the plasmids and grown overnight at 37 °C in LB containing 100 μg/ml ampicillin and 0.5% glucose. Overnight cultures were diluted to $OD_{600} = 0.1$ and grown for ~2 h at 37 °C until the $OD_{600}$ reached 0.8. Expression of the BamA β-barrel was induced by adding 0.8 mM isopropyl β-D-thiogalactoside (IPTG), and the cells were incubated for ~3 h until the $OD_{600}$ reached 3.0. All purification steps were conducted at 4 °C. Cells were collected by centrifugation (4000 x *g*, 20 min, 4 °C), resuspended in PBS pH 7.4

and lysed using Dounce homogenization followed by sonication (VibraCell VCX 750; Sonics). Inclusion bodies were isolated by centrifugation (4000 x $g$, 30 min, 4 °C). After two rounds of washing with PBS pH 7.4, 1% lauryldimethylamine-N-oxide (LDAO; Anatrace), purified inclusion bodies were pelleted by centrifugation (25,000 x $g$, 20 min, 4 °C) and solubilized at 22 °C for 2 h in 20 mM Tris-HCl pH 8.0, 6 M guanidinium-HCl. Refolding was performed by dropwise addition of solubilized BamA into 20 mM Tris-HCl pH 8.0, 0.5% LDAO at 4 °C overnight with vigorous stirring. The final protein concentration (assessed by $A_{280}$) in refolding buffer was 50 μg/ml. Protein purity was assessed to be >90% via SDS-PAGE.

Refolded protein was applied to a Hi-Trap Q Fast Flow column (Cytiva) and eluted with 20 mM Tris pH 8.0, 1 M NaCl, 0.1% LDAO. Peak fractions were pooled and desalted to lower the NaCl concentration to 150 mM for biotinylation reactions using BirA ligase (for proteins that have an N-terminal Avitag) or biotin-maleimide (ThermoFisher) (for the $Ec$BamA' C690S/C700S/S752C mutant). Biotin reaction progression was monitored using LC-MS (Q-ToF, Agilent). Biotinylated $Ec$BamA' was applied to a His-Trap Fast Flow column (Cytiva) equilibrated in 20 mM Tris-HCl pH 8.0, 150 mM NaCl, 0.1% LDAO to remove BirA, and the protein was eluted in the same buffer containing 500 mM imidazole. A Superdex-200 26/600 size-exclusion column (Cytiva) was used for the final purification step in 20 mM Tris-HCl pH 8.0, 150 mM NaCl, 0.1% LDAO. Monodispersed peak fractions were pooled and concentrated to ~2 mg/ml (assessed by $A_{280}$) for biophysical characterization and mRNA display screening.

### mRNA display screening

PDPS was established at Merck & Co., Inc., Rahway, NJ USA under license from PeptiDream (Kawasaki, Kanagawa, Japan). For the first rounds of selection, 100 μL of NNU1 and 100 μL of NNU3 in vitro translation reactions were employed. Partially reconstituted $E. coli$-based in vitro coupled transcription-translation systems were constructed by combining 50 mM HEPES-KOH (pH=7.6), 12 mM magnesium acetate, 100 mM potassium acetate, 2 mM spermidine, 20 mM creatine phosphate, 2 mM DTT, 2 mM ATP, 2 mM GTP, 2 mM CTP, 2 mM UTP, 0.1 mM 10-formyl-5,6,7,8-tetrahydrofolic acid, 0.5 mM of each proteinogenic amino acid Leu, Val, Pro, Tyr, His, Asn, Asp, Arg, Ser, and Gly. The in vitro transcription-translation systems also contained 1.5 mg/mL total $E. coli$ tRNA along with 0.73 μM AlaRS, 0.03 μM ArgRS, 0.38 μM AsnRS, 0.13 μM AspRS, 0.09 μM GlyRS, 0.02 μM HisRS, 0.4 μM IleRS, 0.04 μM LeuRS, 0.16 μM ProRS, 0.04 μM SerRS, 0.02 μM TyrRS, 0.02 μM ValRS, 2.7 μM IF1, 0.4 μM IF2, 1.5 μM IF3, 0.26 μM EF-G, 10 μM EF-Tu, 10 μM EF-Ts, 0.25 μM RF2, 0.17 μM RF3, 0.5 μM RRF, 0.1 μM T7 RNA polymerase, 4 μg/mL creatine kinase, 3 μg/mL myokinase, 0.1 μM pyrophosphatase, 0.1 μM nucleotide-diphosphatase kinase, and 1.2 μM ribosomes. The in vitro transcription-translation system used for the NNU1 library also contained aminoacyl-tRNAs $N$-chloroacetyl-L-phenylalanine (ClAc-F)-tRNA$^{fMet}_{CAU}$, Cys-tRNA$^{Asn}$, MePhe-tRNA$^{Asn}$, MeGly-tRNA$^{Asn}$, MeNle-tRNA$^{Asn}$, and MeAla-tRNA$^{Asn}$, while the in vitro transcription-translation system used for the NNU3 library also contained aminoacyl-tRNAs $N$-chloroacetyl-L-phenylalanine (ClAc-F)-tRNA$^{fMet}_{CAU}$ Cys-tRNA$^{Asn}$, PeGly-tRNA$^{Asn}$, MeGly-tRNA$^{Asn}$, HxGly-tRNA$^{Asn}$, and Tic-tRNA$^{Asn}$. Finally, these reactions each contained 2 μM library mRNA containing the open reading frame [5'-AUG-(NNU)$_{12-13}$-UGGGGAGGUGGUGGAAGUAGCUAG-3'] that was annealed to a puromycin linker (Fig. S1), and a DNA oligonucleotide that was both conjugated to a puromycin moiety at the 5'-end and complementary to the 3'-end of the mRNA. RF1 is omitted from the translation mixture to prevent multiple turnovers and to facilitate covalent attachment of the puromycin moiety to the C-terminal end of the GGGGSS peptide linker. After 30 min at 37° C, the transcription-translation reactions were halted by adding 20 μL 100 mM EDTA, pH 7.5 to dissociate the ribosomes from mRNA-peptide conjugates. Once

liberated from the ribosome, the nascent linear peptides spontaneously undergo macrocyclization[78].

Reverse transcription of the peptide-tethered mRNA was carried out by adding a 40 μL solution containing 200 mM Tris, 300 mM KCl, 72 mM MgCl$_2$, 1.2 mM dNTPs, 12 μM Reverse Transcription Primer (5'-TAGCTACTTCCACCACCTCCCCA-3'), 800 units of M-MLV reverse transcriptase lacking RNase H activity (Promega, catalog# M368A) followed by incubation at 42 °C for 30 min. The reverse transcription reactions were quenched by adding 30 μL 100 mM EDTA, pH 7.5. Macrocyclic peptide-mRNA-cDNA complexes were desalted using gel columns (Micro Bio-Spin P-6 Gel Columns, Bio Rad Cat#732–6221) that were pre-washed with selection buffer (20 mM Tris pH 7.4, 150 mM NaCl, 0.1% LDAO). One microliter of the desalted peptide-mRNA library was saved as input for quantification using quantitative polymerase chain reaction (qPCR). Saturating amounts of chemically biotinylated $Ec$BamA' were immobilized on M-280 magnetic beads (Dynabeads M-280 streptavidin; Thermo-Fisher) by incubating the target protein for 1 h at 4 °C. The maximum loading capacity of the $Ec$BamA' construct on M-280 beads was previously determined during pre-screening optimization. The $Ec$BamA'-bound M-280 beads were washed with selection buffer to remove unbound $Ec$BamA. Panning was carried out by resuspending $Ec$BamA'-bound M-280 beads to a final concentration of 200 nM protein with the desalted peptide-mRNA-cDNA libraries for 1 h at 4 °C with rotation. After incubation, the non-binding peptides were removed by washing the beads three times using selection buffer. The cDNA was eluted from the beads by incubating them at 95 °C for 5 min in polymerase-free PCR buffer (10 mM Tris-HCl pH 8.5, 50 mM KCl, 0.1% Triton X-100, 0.25 mM dNTPs, 2 mM MgCl$_2$, 0.25 μM Forward Primer (5'- CTAGTAATACGACTCACTATAG GGTTAACTTTAAGAAGGAGATATACATATG-3'), and 0.25 μM Reverse Primer (5'- CCCGCCTCCCGCCCCCCGTCCTAGCTACTTCCACCACCTC CCCA-3') followed by collection of the supernatant. The recovered cDNA was quantified by qPCR using 1 μL of the eluted sample. Ten units of Taq polymerase were added to the remaining supernatant and the mixtures were amplified by PCR. Double-stranded DNA was transcribed into mRNA, which were standardized to 20 μM after quantitation using a Bioanalyzer 2100 (Agilent).

In round 2 of the selection, translation reactions were performed as described above except on a 20 μL scale. A preclear step a was incorporated before the panning step to remove potential bead-binding peptides. From round 2 onwards, a Kingfisher™ mL Purification System (ThermoFisher) was programmed to preclear peptide-cDNA-mRNA libraries by incubating them with target-free M-280 beads three times for 10 min before incubating them with $Ec$BamA'-bound beads. To track the isolation of undesired bead-binding peptides, the third set of precleared beads were washed three times, and recovered cDNAs eluted from these beads were analyzed by qPCR. Incubation with the $Ec$BamA' beads for 1 h at 4 °C with agitation and subsequent washes were also performed in a programmed Kingfisher™ instrument. In round 3 and all subsequent rounds, in vitro transcription-translation reactions were performed on a 5 μL scale, dsDNA PCR products from preceding rounds were added instead of mRNA, a puromycin linker was added directly to the reaction mixtures to anneal to in situ-transcribed mRNA, and the incubation of peptide libraries with $Ec$BamA'-bound beads was shortened to 30 min at room temperature with agitation. After six rounds of selection, the percentage of the total cDNA that was recovered from the $Ec$BamA' beads was clearly elevated, which suggested that libraries were enriched in cDNAs that encode $Ec$BamA-binding cyclic peptides. NNU1 and NNU3 post-Round 6 cDNA were sequenced using a MiSeq next-generation sequencer (Illumina).

The peptide pools were assayed for mRNA-free binding activity using plate-based electro chemiluminescence. Post-round 6 dsDNA pools for NNU1 and NNU3 were amplified using primers for appending a FLAG tag followed by a TAA stop codon, 0.25 μM Forward Primer, and

FLAG Reverse Primer (5′-CGAAGCTTACTTGTCGTCGTCGTCCTTG-TAGTCCTTCTTGCTACTTCCACCACCTCCCC-3′) that appends a KKDYKDDDDK tag to the C-terminal end of GGGGSS linker. NNU1 and NNU3 peptide pools were in vitro transcribed and translated on a 5 μL scale for 40 min at 37 °C using modified translation systems that contained an additional 0.2 mM Lys and 0.132 μM LysRS. Due to the stop codon exchange from TAG to TAA that was introduced along with the C-terminal FLAG tag, cyclic peptides were translated in vitro with multiple turnovers. Following termination of each in vitro translation reaction, 85 μL of 20 mM Tris pH 7.4, 150 mM NaCl, 0.1% LDAO were added to dilute the peptide pool. A 96-well High Bind Streptavidin SECTOR Plate (Mesoscale Discovery) was prepared by incubating 3 pmol of biotinylated *Ec*BamA′ in 50 μL BamA buffer (20 mM Tris pH 7.4, 150 mM NaCl, 0.1% LDAO) to one well for each peptide pool that was tested. For negative controls, 3 pmol of biotinylated β-catenin in 50 μL HBS-T (25 mM HEPES pH 7.4, 150 mM NaCl, 0.05% Tween-20) was incubated in one well for each peptide pool that was tested. The biotinylated proteins were incubated on the plate for 60 min with 450 rpm agitation at room temperature. The wells were then washed three times with 150 μL of BamA buffer (for *Ec*BamA samples) or HBS-T (for β-catenin controls) to remove unbound protein. 5 μl HBS-T was added to each 5 μl in vitro translation reaction, and 4 μl of this sample was mixed with 46 μL BamA buffer and added to a *Ec*BamA coated well. Another 4 μl of the sample was mixed with HBS-T and added to the β-catenin coated well. Wells were incubated for 60 min with agitation at 450 rpm at room temperature (~22 °C). After incubation, the wells were washed three times with 150 μL of BamA buffer (for *Ec*BamA samples) or HBS-T (for β-catenin controls) to remove unbound peptide. 50 μL of antibody solution (PBS pH 7.4, 0.1% Tween 20, 1 % BSA, 1:2000 diluted mouse anti-Flag antibody, 1:2000 diluted sulfo-tag-anti-mouse antibody) was added to each target well and incubated for 60 min with agitation at 450 rpm at room temperature. After incubation, the wells were washed three times with 150 μL PBS pH 7.4, 0.1% Tween 20 to remove unbound antibody. After Mesoscale Discovery Read Buffer was diluted with an equivalent amount of ultrapure water to a final concentration of 2x, 150 μL was added to each well and the plate was read in a MESO SECTOR S 600MM instrument (Mesoscale Discovery).

## Peptide synthesis

Peptides CP1-CP3 were synthesized using standard solid phase synthesis methods that employ Fmoc/tert-Bu chemistry[79–82]. Peptide sequences were assembled by solid phase synthesis on a Biotage Syro II (Biotage, Charlotte, NC) synthesizer. Synthesis was started using 25 μmol Rink amide resin, 100–200 mesh, 0.49 mmol/g (ChemImpex). Each amino acid was coupled in 8-fold excess as a 0.4 M solution in dimethylformamide (DMF), which was activated using an 8-fold excess of 0.4 M hexafluorophosphate azabenzotriazole tetramethyl uronium (HATU) in DMF and a 16-fold excess of 2.0 M N,N-diisopropylethylamine (DIEA) in N-methylpyrrolidone (NMP). For Fmoc-deprotection, resin was treated twice with 20% piperidine in DMF (1.5 mL) at room temperature for 3 min. Single and double couplings were performed at 50 °C or 75 °C with 15 min coupling times (see Table S6 for coupling methods and temperatures). Chloroacetylation of the N-terminus was performed by treating the resin twice with a 16-fold excess of chloroacetic anhydride (0.4 M in DMF) and a 32-fold excess of DIEA (2.0 M in NMP) for 10 min at room temperature.

Following solid phase synthesis, the peptidyl resin was treated with 2.0 mL of cleavage solution (trifluoroacetic acid (TFA)/triisopropylsilane/H₂O/dithiothreitol = 94/2.5/2.5/1 (v/v/v/w)) for 1 h at room temperature with shaking on an orbital shaker. The samples were filtered into 50 mL centrifuge tubes filled with 25 mL of chilled diethyl ether. The resin was washed with TFA (0.5 mL) and the TFA was also transferred to the chilled ether. Samples were centrifuged at 3700 x *g*

for 5 min at 4 °C and the supernatant was removed. Additional cold diethyl ether (25 mL) was added to the peptide pellet and the white solid was re-suspended by vortexing. The solid was centrifuged at 3700 x *g* for 5 min at 4 °C, and the supernatant was removed. This process was repeated one more time and the resulting crude peptide pellet was briefly dried under a stream of nitrogen. The crude off-white solids were dissolved in 1.5 mL DMSO. The pH was adjusted by the addition of 2.0 M DIEA in NMP (0.5 mL) to reach pH 8. The solution was shaken at room temperature overnight.

Purification was performed by preparative reversed-phase high-performance liquid chromatography (RP-HPLC) on a Waters™ X-Bridge Prep C18 OBD Prep column (130 Å, 5 pm, column size 19 × 100 mm) using a Waters™ MS-Directed AutoPurification HPLC-MS system. The mobile phase consisted of a mixture of (A) 0.16% TFA in HPLC-grade water and (B) 0.16% TFA in HPLC-grade acetonitrile. Using a flow rate of 25 mL/min, B was gradient ramped from 25% to 50% over 5 min. Purification was monitored via UV at a wavelength of 215 nm.

Confirmation of the identity of final compounds and a purity assessment were performed by UPLC-MS, which was measured by a reverse phase Waters™ ACQUITY UPLC-MS system equipped with a Waters™ XSelect CSH C18 Column (130 Å, 2.5 μm, column size 2.1 × 50 mm). The sample injection volume was 1 μL. The mobile phase consisted of a mixture of (A) 0.05% TFA in HPLC-grade water and (B) 0.05% TFA in HPLC-grade acetonitrile. At a flow rate of 1 mL/min, B was gradient ramped from 5% to 100% over 5 min. Elution was monitored via UV at a wavelength of 215 nm. Lyophilization of combined fractions containing pure peptide resulted in the final cyclized product as a powder. Characterization data for the peptides are shown in Table S5 and the UPLC/MS analyses are shown in Figs. S25–S27.

## SPR assays

For peptide binding assays a Cytiva Series S CAP sensor chip was loaded onto a Biacore T200, and the chip surface was prepared according to the manufacturer's recommendations. The running buffer was 10 mM HEPES pH 7.5, 150 mM NaCl, 0.1% LDAO. Briefly, the CAP reagent (Cytiva) was diluted 1:10 in HBS-EP buffer (0.01 M HEPES pH 7.4, 0.15 M NaCl, 3 mM EDTA, 0.005% v/v Surfactant P20) and flowed across the chip surface for 300 sec. Biotinylated *Ec*BamA, *Pa*BamA, or *Ab*BamA was then immobilized on the surface by flowing the protein over the chip for 200 sec. Cyclic peptides were injected at concentrations of 12.3, 37, 111, 333 and 1000 nM in single-cycle kinetic mode for 120 sec followed by a 120 sec dissociation period, and after the highest concentration of peptide was injected the dissociation period was 1200 sec. The chip surface was regenerated after each peptide was tested by flowing regeneration solution (8 M guanidine HCl, 1 M NaOH) over the chip surface.

SPR competition binding assays were also used to assess the binding of test peptides to the darobactin binding site in *Ec*BamA. Peptide binding affinity was assessed in the absence (intrinsic affinity; $K_{D, intrinsic}$) or presence of 1 μM darobactin (apparent affinity; $K_{D, apparent}$). A significant shift in peptide affinity in the presence of darobactin (weaker $K_D$, apparent versus $K_{D, intrinsic}$) indicates peptide binding to the darobactin binding site. For these assays, the CAP chip preparation and protein conjugation were performed as above. As a control, peptide intrinsic affinity for *Ec*BamA was assessed first. Using single cycle kinetics analysis mode, peptides were sequentially injected at concentrations of 15.625, 31.25, 62.5, 125, 250, 500 and 1000 nM for 110 s at 20 μl/min followed by a wash with running buffer (10 mM HEPES pH 7.4, 150 mM NaCl, 0.1% LDAO, 1% DMSO) for 600 sec. To assess the apparent affinity of peptides for *Ec*BamA in the presence of darobactin, the same steps were followed, except 1 μM darobactin was added to the assay buffer (10 mM HEPES pH 7.4, 150 mM NaCl, 0.1% LDAO, 1 μM darobactin). Data analysis was performed using BIAcore Evaluation software (Cytiva). Binding affinities ($K_D$ values) were estimated by fitting

binding sensorgrams in kinetics mode using a 1:1 binding stoichiometry model.

## Whole cell inhibition assays

The MICs of the cyclic peptides obtained in the mRNA display screens against *E. coli* strains JCM158 and JCM972 were measured using broth microdilution in cation-adjusted Mueller–Hinton broth (CAMHB), as recommended by the Clinical and Laboratory Standards Institute. In addition, zone of inhibition (ZOI) assays using JCM158, JCM972, AM710, AM711, MB5746, *Acinetobacter baumannii* ATCC17987, *Pseudomonas aeruginosa* MB5919, and *Staphylococcus aureus* Col were performed to provide a visual display of the relative susceptibility of the strains to the peptides. Briefly, ZOI studies were performed by first growing the bacteria in CAMHB to late exponential phase at 37 °C. Molten CAMHA (CAMHB containing 1.2% agar) was equilibrated to 48 °C in a water bath for 60 min. Cells were then mixed with 30 mL of the molten CAMHA at a 1:1000 dilution and immediately poured onto an Omni-tray plate (NUNC cat# 242811). The agar plate was kept at room temperature for 20 min to solidify and further dried in a biosafety cabinet for ~20 min. Each peptide was dissolved in DMSO, and 2–5 µL aliquots were dispensed onto the top of the plate in a two-fold dilution series in which the peptides were diluted in DMSO. After the liquid was absorbed, plates were incubated at 37 °C for 18 h, and images were acquired using a Hi-Res imaging system[83].

## Protein expression and purification for X-ray crystallography

*Ec*BamA (422–810) was cloned into pET-23b with a methionine at its N-terminus. *E. coli* BL21(DE3) transformed with the plasmid was grown overnight in 200 mL TB containing 100 µg/ml ampicillin at 37 °C. The overnight culture was diluted into 10 L of the same medium (divided into 10 flasks) and grown to $OD_{600} = 0.8$. Protein expression was then induced by adding 0.8 mM IPTG. After incubating the cultures at 37 °C for 4 h cells were harvested by centrifugation (7500 x *g*, 15 min, 4 °C), and the pellets were frozen in liquid nitrogen and stored at −80 °C.

*Ec*BamA was purified from inclusion bodies by following a previously described protocol[84]. All steps were performed at 4 °C unless otherwise noted. The frozen *E. coli* pellet was resuspended in 5 mL PBS per gram of cells and lysed by Dounce homogenization and sonication. The cell debris and unbroken cells were removed by centrifugation (20 min, 1500 x*g*). The inclusion bodies were pelleted by centrifugation (20 min, 15,000 x *g*) and resuspended in a washing buffer (20 mM Tris-HCl pH 8, 500 mM NaCl, 1% Triton X-100), and Dounce homogenized, pelleted again (20 min, 15,000 x *g*), washed in the same buffer, and then washed in water to remove the Triton X-100. The inclusion bodies were then pelleted at 200,000 x *g* for 20 min. The pellet was air dried, resuspended in a denaturing solution (20 mM Tris-HCl pH 8, 8 M urea), Dounce homogenized, and stirred gently for 2 h. After the solution was centrifuged at 200,000 x *g* for 30 min to remove insoluble material, the supernatant (which contained the denatured *Ec*BamA protein) was dripped into a refolding buffer (20 mM Tris-HCl pH 8, 0.5% LDAO) while stirring at maximum speed (~5 min), and then stirred at a low speed overnight. The refolded protein was loaded onto a Q column and eluted through a NaCl gradient [20 mM Tris-HCl pH 8, 0.6% tetraethylene glycol monooctyl ether (C8E4), 0–1 M NaCl]. The protein peak was further purified by size exclusion chromatography (20 mM Tris-HCl pH 8, 0.6% C8E4, 150 mM NaCl), concentrated to 14.8 mg/ml prior to crystallization, snap frozen in liquid nitrogen and stored at −80 °C.

## X-ray crystallography data collection, refinement, and structure determination

*Ec*BamA-darobactin cocrystals were first obtained through hanging drop vapor diffusion at 18 °C. *Ec*BamA and darobactin were mixed with the crystallization solution [6% v/v Tacsimate pH 6, 0.1 M MES pH 6, 2.5% w/v tetrabutylphosphonium bromide (TBPB), 7–10% w/v PEG 4000] at a 1:1 v/v ratio. The cocrystals were then transferred into a 2 µL soaking drop reservoir supplemented with 1.2% w/v C8E4 and either 1 mM CP1 or 5 mM CP2 to soak in the peptides. The crystals were soaked at 18 °C with CP1 or CP2 for 1 day and 8 h, respectively, prior to flash cooling in liquid nitrogen.

CP3 was co-crystallized with *Ec*BamA through sitting drop vapor diffusion at 18 °C. The protein (14.8 mg/mL) was pre-incubated with 2.5% w/v TBPB and 0.5 mM CP3 for 30 min on ice and then mixed with crystallization solution (0.2 M sodium formate, 20% PEG3350) at a 1:1 v/v ratio. Crystals were looped directly from the drops and flash cooled in liquid nitrogen.

Diffraction data for CP1 and CP2 soaked crystals were collected at the Canadian Light Source (CMCF-ID beamline), while data for CP3 cocrystals were obtained at the Swiss Light Source (X06SA-PXI beamline). Details are shown in Table S1. All datasets were processed using the Global Phasing autoPROC pipeline[85–89] and the STARANISO[90] output was used for molecular replacement and structural refinement. Phasing by molecular replacement was performed using CCP4 Dimple[88], with an *Ec*BamA-darobactin complex structure or an apo-*Ec*BamA structure as a searching model. Peptide compounds were built manually in Coot[91]. The compound and covalent linkage restraints were generated using the Global Phasing tools Grade[92] and aB_covalent[93], respectively. Refinement was conducted using the Global Phasing Buster package[93]. Final coordinates and structure factors were deposited into the RCSB Protein Data Bank under the codes 9CS0, 9CS1, and 9CS2 for BamA bound to darobactin and CP1, BamA bound to darobactin and CP2, and BamA bound to CP3, respectively.

## All-atom MD simulations

We conducted all-atom MD simulations to gain insights into the dynamics of CP1 and CP2 binding to BamA. First, we used the structures of CP1 and CP2 bound to the closed state of BamA to assess the stability of the lumen-binding peptides over the course of the simulation. To further elucidate the mechanism by which the peptides inhibit the activity of BamA, we also docked the two peptides into the BamA β-barrel in an open state (PDB: 5LJO chain C) by aligning the backbone of the open state BamA β-barrel with the backbone of the closed state and relaxed the starting docked state. A six step equilibration totaling 1.875 ns was performed independently for each replica. We performed additional MD simulations to assess the influence of the peptides on the open state of the BamA β-barrel. The simulation system information is summarized in Table S7.

To mimic an *E. coli* OM, we built an asymmetric bilayer using a Gram-Negative Outer Membrane Modeler (GNOMM)[94]. The outer leaflet of the membrane was composed entirely of lipid A, and the inner leaflet consisted of a mixture of POPE and POPG in a 7:3 ratio. Subsequently, we inserted the crystal structure into the bilayer using VMD[95]. $Ca^{2+}$ ions were added to lipid A to neutralize each system, and 0.15 M NaCl was also added to the bulk solution in GROMACS[96]. The TIP3P water model was used in our simulation systems[97]. All systems were relaxed prior to the start of the simulations.

All of the simulations were performed with GROMACS. The production run was performed for 50 ns. The time step was set to 2 fs, and we constrained bonds containing hydrogen atoms with the SHAKE algorithm[98]. Van der Waals interactions were smoothly switched off between 10 and 12 Å by the force-based switching function[99], and the long-range electrostatic interactions were calculated by the particle-mesh Ewald method[100]. The CHARMM36 force field[101] was used for all the simulations. The simulation was performed for each system using the NPT (constant pressure and temperature) at 1 bar and 310 K. All the results were based on the analysis of production simulations.

RMSDs of the peptide backbones were calculated to assess their stability during the MD simulations. Prior to the RMSD calculations, the protein structure was aligned to its initial crystal structure based on the protein backbone. The pore profile was calculated using the HOLE2

program[102]. Hole radius calculation for BamA was performed with the Z coordinate of the center of mass (COM) of the membrane set to 0. The obtained pore radius data along the z-axis were averaged and plotted using Matplotlib[103].

## Analysis of OMP assembly in vitro and in living cells

The assembly of OMPs in vitro was analyzed using a previously described assay[62]. EspPΔ5′, OmpT and OmpC were synthesized using the PURExpress coupled transcription-translation system (New England Biolabs) according to the manufacturer's instructions, except that reactions were supplemented with 2 μM purified SurA and sonicated proteoliposomes containing purified BAM (0.5 μM) and 1-palmitoyl-2-oleoyl-glycero-3-phosphocholine (POPC). Typical 10 μL reactions also contained murine RNAse Inhibitor (8 U) and 0.4 μM FluoroTect GreenLys (Promega), a lysine-charged tRNA labeled with the fluorophore BODIPY-FL at the ε position that was used to introduce fluorescent lysine residues into each OMP during translation. OmpC assembly reactions also contained 0.5 mM Rc-LPS (Millipore Sigma) prepared as previously described[63]. After all of the above reagents were added to a master mix and aliquoted, the samples were mixed with a cyclic peptide, darobactin or Polyphor 7 diluted to the appropriate concentration in 20 mM Tris pH 8.0 (or the buffer alone) and incubated at 30 °C for 5 min in a Thermomixer R (Eppendorf) with shaking (600 rpm). Subsequently a pET28b-based plasmid that encoded an OMP[62] was added to each reaction to a final concentration of 10 ng/mL and samples were incubated at 30 °C for an additional 30 min with shaking. Reactions were then stopped by placing the tubes on ice and adding RNaseA (0.5 mg/ml). In some OmpC assembly experiments, the reactions were divided in half, and one half was treated with 30 μg/mL proteinase K (PK) on ice for 15 min. Phenylmethylsulfonyl fluoride (PMSF; 10 mM) was then added to stop the protease digestion. At the end of all experiments one half of each sample was heated to 95 °C for 5 min while the other half was left on ice. Proteins were then resolved by SDS-PAGE on 8–16% Novex Tris-glycine minigels (ThermoFisher) and visualized using a Fuji FLA 9000 imager. For quantitation (which was performed using Fuji MultiGauge v. 3.1 software), the fraction of each OMP that was assembled in the absence of a cyclic peptide was defined as 100%.

To analyze the assembly of OMPs in living cells, overnight cultures of HDB164 (MC4100 *ara^R bamA101 ompT::spc*) transformed with pJH110 (P$_{trc}$-*espPΔ5*)[65] or AD202 transformed with pRSL5 (P$_{trc}$-*espP*)[77] or pJH224 [P$_{trc}$-*espP* (G1123R)][67] were grown in M9 supplemented with 100 μg/mL ampicillin (M9/ampicillin) at 37 °C. The overnight cultures were washed in M9 and added to 50 ml fresh M9/ampicillin at OD$_{550}$ = 0.02. When the cultures reached OD$_{550}$ = 0.2, IPTG (10 μM) was added and the cells were incubated for another 30 min at 37 °C before being subjected to pulse-chase labeling as previously described[66]. At each timepoint 1 ml aliquots were removed and proteins were precipitated by adding 10% (v/v) TCA. In most experiments, cyclic peptides dissolved in DMSO were added at 2x MIC either 5 min before or 2 min after the start of the chase. As a control, an equal amount of DMSO was added. In some experiments, 2.5 mM EDTA, pH 8 was added instead of a cyclic peptide. For experiments that involved an analysis of the effect of sorbitol on the assembly of EspP(G1123R), 5.5 ml aliquots were subjected to pulse-chase labeling. After 1 ml was TCA precipitated at the 0 min timepoint, the remaining 4.5 ml was incubated for another 2 min at 37 °C and pipetted over 2 ml M9 pre-frozen in 15 mL tubes. Cells were then centrifuged (3000 x *g*, 10 min, 4 °C), resuspended in the same volume of M9 containing 0.8 M sorbitol or M9, and returned to the 37 °C shaking water bath. The time at which the 37 °C incubation was restarted was defined as 2 min post-chase. The same procedure was used for control experiments that involved an analysis of the effect of sorbitol on the assembly of wild-type EspP,

except that 1 ml was TCA precipitated at the 1 min timepoint, and the remaining 4.5 ml was incubated for another 1 min at 37 °C. After pipetting the cells over 2 mL frozen M9 and centrifugation cells were resuspended in the same volume of M9 containing 0.8 M sorbitol or M9, and then placed in a 28 °C shaking water bath. The time at which the 28 °C incubation was started was defined as 2 min post-chase. To examine the status of the EspP(G1123R) β-barrel after it binds to BamA, 1 ml aliquots of radiolabeled cells were removed and pipetted over ice at each timepoint. Cells were pelleted at 5000 x *g* for 6 min, resuspended in 1 ml 10 mM Tris pH 8.0/40% sucrose, and treated with 100 μg/ml lysozyme/2 mM EDTA for 20 min on ice. The samples were divided in half, and one half was TCA precipitated. The other half was incubated with 200 μg/ml PK on ice for 20 min and then 2 mM PMSF for an additional 15 min prior to TCA precipitation. Immunoprecipitations were performed as previously described[104], and proteins were resolved by SDS-PAGE as described above. Radioactive proteins were visualized on a Fuji FLA 9000 imager using phoshophorimaging settings, and data was quantitated using Fuji MultiGauge v. 3.1 software.

For experiments that involved EspP or EspP derivatives, the accumulated β-barrel was defined as the normalized β-barrel signal, where the normalization factor was (preproEspP$_{signal}$ + proEspP$_{signal}$ + β-domain$_{signal}$) at the 0 min timepoint for each condition and each experiment. For some experiments that involved EspP or EspP derivatives, the percent cleavage at each timepoint was also measured and was defined as [β-barrel $_{signal}$ /(proEspP$_{signal}$ + β-barrel$_{signal}$)] x 100. To account for the loss of signal due to the release of the full-length passenger domain, the β-barrel signal was multiplied by 3.14. For experiments that involved OmpC and OmpA, percent remaining at each timepoint was defined as (signal/signal at 0 min timepoint) x 100, and for experiments that involved wild-type EspP or the EspP(G1123R) mutant the percent remaining was defined as (signal at 30 min timepoint/signal at 5 min timepoint) x 100. Because the 1300 residue EspP protein cannot be completely synthesized within the pulse labeling period, the radioactive signal continues to increase until ~2 min into the chase[77,105].

Site-specific crosslinking experiments were performed using a previously described protocol[66]. Overnight cultures of AD202 transformed with a derivative of pRI22 that contains the EspP(G1123R) mutation[67] and pDULE[106] were grown in M9 supplemented with 100 μg/mL ampicillin and 5 μg/mL tetracycline. After cells were washed they were added to 50 ml M9 at OD$_{550}$ = 0.03. When the cultures reached OD$_{550}$ = 0.2, 4-benzoyl-L-phenylalanine (Bpa; 1 mM) and IPTG (10 μM) were added and the cells were incubated for another 30 min at 37 °C before being subjected to pulse-chase labeling. At each timepoint 2 mL were removed from the cultures and pipetted into 15 mL tubes over ice, and 4 mL were added to ice in 6 well Falcon tissue culture plates and irradiated on ice for 4 min using a Spectroline SB-100P high intensity lamp. Cells were centrifuged (3500 x *g*, 10 min, 4 °C), resuspended in 1 mL M9 salts, and proteins were TCA precipitated. Proteins immunoprecipitated by anti-EspP and anti-BamD antisera were then resolved by SDS-PAGE and visualized as described above.

## Dynamic light scattering (DLS)

All dilutions were performed in 20 mM Tris pH 8 that was passed through a 0.22 μM filter. Samples (10 μL) containing 0.5 mM BAM-POPC and an appropriate concentration of a cyclic peptide or darobactin were incubated at 30 °C for 10 min. The samples were then placed on ice and pipetted into Wyatt WNDMC microcuvettes immediately before data was collected using a DynaPro NanoStar DLS detector (Wyatt Technology). The parameters for the DLS experiments were as follows: acquisition time, 5 sec; read interval, 1 sec; DLS number acquisition, 10; measurement type, molar mass and size; set temperature, 20 °C; auto-attenuation, on.

 

## Detection of LPS released from *E. coli*

Overnight cultures of AD202 transformed with pJH224 were grown at 37 °C in M9/ampicillin, washed, and added to 55 ml of fresh medium at $OD_{550} = 0.02$. When cultures reached $OD_{550} = 0.2$, 10 µM IPTG was added and cultures were incubated at 37 °C for an additional 15 min. Cultures were then divided into five 10 ml aliquots and placed into 15 ml tubes. Cells were pelleted in 15 ml tubes (3000 x $g$, 15 min, room temperature), resuspended in 1 ml pre-warmed M9 and transferred to 50 ml tubes. At that point EDTA (2.5 mM), a cyclic peptide (2x MIC), or DMSO was added, and samples were returned to the shaking water bath and incubated at 37 °C for 5 min or 15 min. Cells were then placed into 1.7 ml tubes on ice for 5 min and centrifuged (2900 x $g$, 6 min, 4 °C). The cell pellet was discarded and the supernatant was centrifuged in a TLA.2 rotor in a Beckman Optima MAX-TL ultracentrifuge (100,000 x $g$, 30 min, 4 °C). Once again the pellet was discarded, and 5 mL 4x LDS sample buffer was then added to 20 mL of each supernatant (which represents the culture medium of ~0.05 OD equivalents), and molecules located in the culture medium were resolved by SDS-PAGE on 12% Novex Tris-Bis minigels (ThermoFisher) using MES running buffer. LPS was visualized using the Molecular Probes Pro-Q Emerald 300 Lipopolysaccharide Gel Stain Kit (ThermoFisher) following the manufacturer's instructions.

## Reporting summary

Further information on research design is available in the Nature Portfolio Reporting Summary linked to this article.

## Data availability

The final coordinates of BamA bound to darobactin and CP1, BamA bound to darobactin and CP2, and BamA bound to CP3 have been deposited in the RCSB Protein Data Bank with accession codes 9CS0, 9CS1, and 9CS2, respectively. The protein structures that we refer to can be accessed through the PDB using the following URLs: 7NRE and 5LJO. Source data for the experiments performed in this study are provided as a Source Data file.

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

## Acknowledgements

Part of the research described in this paper was performed using beamline CMCF-ID at the Canadian Light Source, a national research facility of the University of Saskatchewan, which is supported by the Canada Foundation for Innovation (CFI), the Natural Sciences and Engineering Research Council (NSERC), the National Research Council (NRC), the Canadian Institutes of Health Research (CIHR), the Government of Saskatchewan, and the University of Saskatchewan. We acknowledge the Paul Scherrer Institut, Villigen, Switzerland for provision of synchrotron radiation beamtime at beamline X06SA-PXI of the SLS. We thank Barry Cunningham, Tiuana Howard, Natalya Pissarnitski, Michael Yang, Valerio Piacenti, and Federica Orvieto for their assistance synthesizing and purifying peptides. We thank Petr Vachal for helpful discussions and guidance on peptide chemistry. We would also like to thank Di Wu and Greg Piszczek (Biophysical Core Facility, National Heart, Lung, and Blood Institute) for providing help with the DLS experiments, Matt Doyle for providing insightful comments on the manuscript, and Ann Marie Norris for help drawing Fig. 7. The research performed by J.H.P. and H.D.B. was supported by the Intramural Research Program of the National Institute of Diabetes and Digestive and Kidney Diseases. Funding for all other authors was provided by Merck Sharp & Dohme LLC, a subsidiary of Merck & Co., Inc., Rahway, NJ, USA.

## Author contributions

S.L., R.E.P., D.K., A. Walji, A. Weinglass, T.M.K., A.S., J.S., H.D.B., and S.S.W. conceptualized the experiments and supervised the study. T.M., Y.H., and J.L. performed protein purifications, W.Z. and C.H. performed and analyzed mRNA display screens, J.F. synthesized peptides, H.W. performed and analyzed MIC assays, J.P. and A.S. performed and analyzed SPR experiments, M.E.W., M.G., J.L., and D.K. performed and analyzed X-ray crystallography experiments, and H.Z., S.M-V., and J.J. performed and analyzed MD simulations and molecular modeling studies. J.H.P. and H.D.B. performed functional studies to determine the effect of the cyclic peptides on the function of BamA in vitro and in live cells and analyzed the data. M.E.W., H.Z., C.H., and H.D.B. wrote the manuscript. All authors reviewed and edited the final manuscript.

## Competing interests

Authors M.E.W., W.Z., H.W., J.P., A.S., H.Z., T.M., Y.H., S.M-V., J.F., J.L., J.J., C.H., S.L., R.E.P., D.K., A. Walji, A. Weinglass, T.M.K., A.S., J.S., and S.S.W. are or were employees of Merck Sharp & Dohme LLC, a subsidiary of Merck & Co. Inc., and may hold stock or stock options in Merck & Co,. Inc., Rahway, NJ, USA. Authors J.H.P., H.D.B., and M.G. declare no competing interests.
