## [Transparent Peer Review file · Nature Communications]

Antibacterial macrocyclic peptides reveal a novel mode of BamA inhibition

Corresponding Author: Dr Morgan Walker

Version 0:

Reviewer comments:

Reviewer #1

(Remarks to the Author)

Folding and insertion of beta-barrel proteins into the gram-negative outer membrane is an essential process carried out by the beta-barrel assembly machine (BAM). BamA, the central and conserved component of the BAM complex, is a potential antibiotic target, however most recently described inhibitors have key limitations. In their manuscript Antibacterial macrocyclic peptides reveal a novel mode of BamA inhibition, Dr. Harris Bernstein, Dr. Scott Walker, and co-workers describe the discovery and characterization of three novel BamA-targeting macrocyclic peptides. Though these peptides fall short in their ability to inhibit growth of wild-type bacteria, they are enabling in that they identify a previously unappreciated inhibitory binding sites in the BamA lumen, display antibacterial activity against sensitized strains, and enable important characterization of outer membrane protein biogenesis. Overall, these peptides provide compelling insights that inform the mechanism of BAM and future antibiotic discovery and optimization.

The experiments presented in the paper are well-designed, thoroughly justified, and logical. The manuscript is very well written, and the conclusions drawn by the authors are justified by the clearly presented data. The exciting results presented in this work reveal important insights that will be of high interest to broad readership of Nature Communications. Though I have no major concerns about this manuscript in its current state, I have provided some questions and comments below that could help clarify some points or, in some cases, extend the findings.

- Was there a rationale for choosing the non-natural amino acids that were included in the NNU1 and NNU3 libraries you generated?
- How many different peptides were encoded in the library after the 6th round of enrichment and what were the criteria used to choose CP1, CP2, and CP3 for follow up characterization? Were there any variants (e.g., substituted amino acids at single positions) of CP1, CP2, or CP3 present in the enriched libraries of BamA binders? This could provide extremely valuable insight for any future efforts to optimize these macrocycles.
- The binding spectrum of CP1, CP2, and CP3 provides useful information. Was binding to BamA from other Enterobacteriaceae species assessed (e.g., *Klebsiella pneumoniae*)? Are you able to rationalize the selective binding to *E. coli* BamA over *P. aeruginosa* BamA and *A. baumannii* BamA by the amino acid identities at the structurally determined macrocycle binding sites? This could point to interactions that could be the focus of future efforts to broaden their spectrum.
- Did you evaluate binding of CP1, CP2, and CP3 to the BAM complex (and not just BamA) by SPR? Given the fact that reported structures of BamA differ at the lateral gate based on the presence of the accessory lipoprotein, it could provide some insight into binding (and could explain the diminished activity you observe with the strains lacking BamB).
- Have you tested the whole cell activity of CP1, CP2, and CP3 against *E. coli* strains with a compromised outer membrane barrier (e.g., *imp4213* mutant) or in the presence of outer membrane permeabilizing agents (e.g., polymyxin B nonapeptide)? This could provide insight into the points raised about how these peptides access their target within the BamA lumen. You should also consider testing these against efflux defective strains.
- There are some additional useful controls to add for the MIC testing. First, MICs against *P. aeruginosa* and *A. baumannii* as negative controls (you demonstrated that the macrocycles do not bind to their BamA). Second, an MIC against a gram-

positive strain would identify any non-specific activity (or this could be assessed in a mammalian cell assay). Third, with wild-type *E. coli*, even though they do not kill the bacteria, it could be enabling to measure if macrocycle inhibition of BamA is sufficient to potentiate the activity of antibiotics that are normally excluded by the intact outer membrane. Finally, given the observations with darobactin, I am wondering if you considered looking at whether there is an additive, synergistic, or even antagonistic effect if you co-treat with darobactin and one of your macrocycles.

- Have you attempted to select for mutants resistant to CP1, CP2, or CP3? These selections, even with the sensitized strains, could be extremely informative to provide additional insight into binding, identify any possible bypass resistance mechanisms, further differentiate the three macrocycles, and illuminate BamA function.

- Not suggesting adding more structures (already a ton of useful information in the manuscript), but did you consider looking at the structures of BamA with just CP2? Also, how about the structures of any of the macrocycles with the full BAM complex instead of just BamA? I guess it is not entirely clear why you chose to solve the structures of BamA with CP1 and CP2 in the presence of darobactin and CP3 in the absence.

- The *in vivo* folding assays are super informative. Can you speak to the degree of OMP folding inhibition and the impact on growth of *E. coli*? Presumably the lack of complete inhibition by the macrocycles accounts for their inability to kill wild-type *E. coli* but this is still sufficient to kill the sensitized strains. Does this provide insight on exactly how much BamA inhibition *E. coli* can tolerate?

- The analysis of the BamA lateral gate hole radius is quite interesting. Curious about whether you can model substrates in this as well? Also, it looks like darobactin was excluded in the MD simulations with CP1 and CP2 (Fig. 3A and 3B). Is this correct? Would you predict that the presence of darobactin would have any impact? Visually, I found it a little difficult to compare across the traces for the different replicates and conditions (for example in Fig. 3c and 3d) and thought it might be helpful to have some overlay versions (even if only in the supplemental data) to better illustrate the comparisons.

- Regarding the 'inaccurate initial docking position of the peptide' (CP1 hole radii experiments at the bottom of page 16), how many different starting positions did you explore? Could other starting positions provide a different picture of what is going on here?

Minor:

- In Fig. 1B, the macrocyclization bond appears to be off in CP2 and CP3 cartoons (at least in the printout I made).

- On page 16, for the CP1 hole radii, it refers to Fig 3C and S22, but I don't think the CP1 results are in 3C (unless it is referring to 3C for the apo...in which case, can you replicate the apo data in S22 to allow for an easier direct comparison).

- What is the blue structure in Fig. S10A (I think CP1 is pink and BamA is grey, but completely missed what the blue structure is)?

Reviewer #2

(Remarks to the Author)

Antibacterial macrocyclic peptides reveal a novel mode of BamA inhibition

Summary/key results

The manuscript focuses on the discovery of novel cyclic peptides that inhibit the essential β -barrel assembly machinery (BAM) in *E. coli*, specifically targeting its central component, BamA. BAM is responsible for the folding and membrane insertion of outer membrane proteins (OMPs), and BamA is a key antibacterial target in Gram-negative bacteria due to its conserved structure and critical function. Using mRNA display, the researchers identified three monocyclic peptides (CP1-3), from a library of 1012 unique cyclic peptides, that bind with high affinity to BamA, inhibiting bacterial growth in *E. coli* strains with BAM deficiencies, and OMP assembly of several model *E. coli* proteins both *in vitro* and *in vivo*. Remarkably, unlike previously characterized inhibitors, such as darobactin, these cyclic peptides bind to unique sites within the BamA β -barrel lumen. The crystal structures show that CP1 and CP3 lock the gate by binding to both sides of the BamA β -barrel, while CP2 restricts its flexibility. MD simulations confirmed the stable binding of these peptides. Taken together, these results revealed a novel mechanism of inhibition. Intriguingly, these peptides also accelerate the assembly of a mutant OMP when added after it binds to BamA, suggesting that the peptides stabilize BamA in a conformation that blocks early stages of OMP assembly but facilitates later steps. This work highlights a new potential approach for antibacterial drug development by inhibiting BamA to control OMP assembly.

Validity

Good.

The work presented is well executed and will be of interest to readers of Nature Communications. Before publication, a few minor formalities need to be addressed.

The interpretation of the data in this study appears valid and is supported by a series of complementary experiments (binding assays, MIC assays, crystal structures, MD simulations and assays to evaluate the inhibition of OMP assembly *in vitro* and *in vivo*). The conclusions regarding the novel mode of inhibition of BAM by cyclic peptides are well-founded, as they are consistently supported by the experimental data. However, although the results are convincing, the high concentrations of peptides required for activity could limit their immediate therapeutic applicability, which the authors

acknowledge, and it is not the main focus of the study. Further exploration of the specificity of peptides for BamA, such as proteomic studies, would strengthen and broaden the impact of the study's findings. This new approach could also inspire further exploration of cyclic peptides or small molecules that target essential bacterial pathways by similar mechanisms.

Significance

High

This manuscript presents the discovery of a new mode of inhibition of BamA, a promising target for the development of new therapeutic strategies against Gram-negative bacterial infections. By identifying cyclic peptides that bind to BamA, the study reveals a new mechanism distinct from known inhibitors such as darobactin. These peptides uniquely interact with sites in the β -barrel lumen of BamA, blocking early steps in outer membrane protein (OMP) assembly while accelerating later steps under certain conditions. Although high peptide concentrations are required for activity, the research focuses on the novel binding mechanism. The discovery made by the authors is important and will have consequences in the field of BamA inhibitors.

Data and methodology

Good.

The manuscript is well-structured, with a logical progression from background to results and conclusions. The use of the mRNA display platform to generate a large peptide library is well justified, focusing on BamA in *E. coli* to target a key bacterial pathway. While the antibacterial spectrum is narrower compared to darobactin, the study introduces a novel mode of BAM inhibition, adding a unique aspect to the research. The approach is solid and the assays that were chosen for this article to prove a new mechanism of interaction with BamA are relevant.

Figures are clear and informative. The data presented are of good quality, with sufficient controls (darobactin, just DMSO...). However, there is no mention of how many replicates were performed for each experiment (Figures 4, 5 and 6). Add the gels and data related to the replicates in the SI. Moreover, additional dose-response data would be useful to better assess the concentration-dependent effects of peptides in various assays, especially for the assay Figure 4F (1 or 2 points after 2 μ M). The subsequent analysis of the results is generally good but could benefit from additional clarification and a more balanced interpretation.

Please find below my remarks and questions for the different parts of the paper.

Abstract

Page 1. You wrote: ". We identified peptides that arrest *E. coli* growth,". However, the cyclic peptides were antibacterial when added to *E. coli* strains that have BAM deficiencies. Please re-phrase.

Page 1. Please replace "in vivo and in vitro" with "in vitro and in vivo" and check the usage of "in vivo" vs "whole-cell" (c.f. comment to page 8)

It may be worth including a short explanation on the source of the cyclic peptides in the abstract.

Introduction

• Page 4. You describe briefly some known BamA inhibitors. Please add other cyclic peptides obtained by synthesis, such as JN-95. This part could be described in more detail to show that there are not many effective darobactin inhibitors already described, particularly peptide inhibitors.

• The darobactin is a bicyclic molecule, so why did you decide to focus only on monocyclic synthesis? It would be also interesting to have bicyclic peptides in this study to mimic darobactin. Page 5 : "Given that darobactin is based on a bicyclic peptide scaffold, we reasoned that small, constrained cyclic peptides might be the most suitable modality for BamA inhibition." Can your peptides be described as small and constrained?

A related question is whether it should be made more explicit that the cyclic peptides were initially designed to mimic darobactin. As a result, the new binding mode is surprising.c

Results and discussion

• Page 6. Were only 3 peptides from the large library shown to bind the *E. coli* BamA β -barrel in vitro? If not, could you briefly explain how these 3 peptides were selected?

• Page 7-Figure 1. The MIC was determined only on *E. coli*. Why did you choose only 3 Gram-negative strains for BamA binding? Given that they bind only to *E. coli* BamA, tests on other strains, like the strains mentioned in the introduction (*K. pneumoniae*, *Enterobacter* spp) would be interesting? BamA binders are supposed to be active on only Gram-negative bacteria. As a result, it would be important to confirm the lack of activity against Gram-positive bacteria (as a negative control because Gram-positive do not possess the BAM complex and they are not affected by the darobactins). Indeed, on page 4 in the introduction, a drawback of MRL-494 is mentioned as it exhibits off-target activity against Gram-positive bacteria.

• Page 7. The peptides interact with BamA but the MIC values on *E. coli* WT are very high. Can you explain that? Do you think that the different position in the BamA pocket can explain the high MIC values?

Please Add the MIC values on AM710 (MC4100 Δ bamB36) and AM711, on Figure 1D.

• Page 7. Did you test the affinity of the peptides to LptD which is also an OMP and essential for viability, and more largely different transmembrane proteins? For example the peptide JB-95 targets BamA and LptD (Urfer, M.; et al., A peptidomimetic antibiotic targets outer membrane proteins and disrupts selectively the outer membrane in *Escherichia coli*. *J. Biol. Chem.* 2016, 291, 1921–1932.)

• Page 7. You wrote ". The activity of the peptides against these mutant strains suggests that they not only bind to BamA, corroborating the SPR binding data, but also impair its function." Did you consider to perform proteomics studies, like Photo-affinity interaction mapping, to evaluate the impact of these new peptides on proteins?

• Page 7. You wrote "Interestingly, the peptides did not bind to either *P. aeruginosa* or *A. baumannii* BamA β -barrels (Figure S5) and therefore lack the broad-spectrum activity of darobactin." Please provide a possible explanation?

- Page 8: “and as it is when darobactin binds to BamA in vivo⁵⁵.” I would recommend to carefully review the use of the term “in vivo” as it seems to be always used when whole-cell experiments with bacteria are meant. In drug discovery, “in vivo” is normally reserved for experiments in animal models. This may confuse the readers.
- Page 8. “Because darobactin does not affect the ability of the cyclic peptides to bind tightly to BamA”. It may be more prudent to write: darobactin does not appear to affect the ability to...
- Page 8. For the crystallisation experiments with CP1 and CP2, BamA was used in complex with darobactin for soaking. Why was the same protocol not also used for CP3?
- Page 10. “Cyclic peptides identified by mRNA display strongly inhibit BAM activity in vitro and in vivo”. They did not strongly inhibit BAM activity in vivo, please change the phrasing and also reconsider the use of “in vivo”. Moreover, you wrote page 12 “Consistent with our in vitro data, we found that CP1, CP2 and CP3 also inhibit BAM activity in vivo, but less effectively.”
- Page 12. For the experiment with living bacteria, please confirm whether the solubility, cytotoxicity and the stability of the peptides was determined before running these studies.
- Page 13 and discussion part. Could an experiment be designed and run to validate the hypothesis concerning the difference in BAM inhibition in vitro vs in vivo?

SI

The SI is well documented, with figures and tables that support and reinforce the data presented in the paper. However, we do not have any information regarding the analysis of these 3 peptides. Please can you add at least the yield, HRMS and the purity of these compounds (chromatogram or elemental analysis).

Figure S19: please add the time of irradiation by UV.

Suggested improvements

See data and methodology for my remarks/questions.

Clarity and context

Overall, the manuscript is clear and well-structured, with research objectives and results presented in a logical manner. The introduction provides sufficient context to understand the importance of BamA as a target for Gram-negative bacterial infections and the use of cyclic peptides for antibacterial purposes. While the technical content is clear for an expert audience, certain sections (particularly the mechanistic details of OMP assembly) could benefit an additional explanation to improve accessibility to a broader readership, and perhaps need a scheme to aid understanding of the experiment.

References

The manuscript appears to be well-referenced, with appropriate references to foundational and recent studies in the field of OMP assembly, BAM inhibition, and BamA inhibitors. Nevertheless, if there are any emerging studies or reviews that provide more comprehensive insights into mRNA display techniques or recent developments in antimicrobial peptide research, including them would further strengthen the context.

Finally, you can add the review “Exploring the Darobactin Class of Antibiotics: A Comprehensive Review from Discovery to Recent Advancements”, *ACS Infect. Dis.* 2024, 10, 2584-2599”

Your expertise

The last part of the paper, on the study of how peptides accelerate a later stage assembly, is outside of our expertise. This part was difficult to understand and may need a scheme to improve comprehension of the experiment.

Reviewer #4

(Remarks to the Author)

In this manuscript, Walker et al. implement an mRNA display approach to successfully select for cyclic peptides that selectively bind with low nanomolar affinity to *E. coli* BamA, the central component of the essential beta-barrel assembly machine (BAM). Unfortunately, the peptides did not display significant antibacterial activity against wild type *E. coli*, but the authors do show activity against *E. coli* mutants with impaired BAM function at micromolar concentrations. They further show that a mutant of BamA (E470K) suppressed the antibacterial activity of two of the peptides, validating BamA as the in vivo target for activity.

The cyclic peptides do not compete with darobactin, a competitive inhibitor of BAM that binds to the beta1 strand of BamA, suggesting a different mechanism of action. This is validated crystallographically, revealing BamA intra-lumen binding sites for all three peptides and providing a rationale for the suppression effect of the BamA E470K mutation. Furthermore, the authors clearly demonstrate BAM inhibition by all three target peptides both in vitro and in vivo using multiple model proteins.

Taken together, excellent data very strong support the conclusion that the selected cyclic peptides inhibit BAM-mediated OMP assembly by an “allosteric” mechanism: binding to the lumen of the BamA barrel that disfavor opening of the “lateral gate” required for full engagement of the substrate OMP. For CP1 and CP3 this appears to be due to binding across the gate, whereas CP2 may constrain the flexibility required for gate opening. Given the luminal binding sites for the peptides, I agree with the author’s interpretation that the low antibacterial activity despite the high binding affinity likely reflects poor periplasmic availability or impaired access to the lumen through the extracellular loops of BamA.

The authors then go on to study the effect of the cyclic peptides in the in vivo assembly of the slow folding mutant EspP(G1123R). Complete folding and assembly of EspP result in cleavage and release of the beta domain. They show that, as expected, addition of the peptides to the cultures before the chase step result in assembly inhibition. However, the observed effect of addition of peptides 2 minutes after the chase is interpreted as evidence of assembly acceleration. This interpretation is predicated on the quantification of the beta-domain bands as well as the preproEspP(G1123R) and proEspP(G1123R) bands. The authors then report the “percentage cleaved”, apparently normalized to the total for each time point. From visual evaluation of the gels, it seems that the main difference between the controls and the “2’ post-chase” conditions is the reduction in intensity of the ProEspP(G1123R) band, not increase of the beta domain. This could be due to faster degradation of the preEspP(G1123R) precursor in the presence of the peptides, particularly if there is a fraction that never folds. It is likely that addition of 25 μ M peptides would trigger the stress response with upregulation of protease activity in the periplasm, which may be responsible for the degradation of preEspP(G1123R). As the experiment seems to be evaluating the BAM activity under single turnover conditions for the labeled protein, quantification should be done normalizing the amounts of beta domain at each time point to the time zero amounts of preproEspP(G1123R) and proEspP(G1123R) bands. The time zero is identical for the control and the “2’ post-chase” condition and therefore appropriate for normalization. The authors observations may still hold true, but the suggested normalization would account for the possibility of degradation of a non-folding (or perhaps released from BAM) EspP(G1123R) fraction.

The evaluation of molecular dynamics experiments where CP2 was docked to the open conformation of BamA and compared to the apo-open and apo-closed conformations has several issues to be addressed. 50ns may be too short a simulation to study the desired effects. More than one replica of the apo-open trajectory should be considered and the closed state should also be relaxed to see if the system simply drifts to similar final state regardless of the starting apo state. Crucially, there is no statistical test to evaluate whether the differences are significant. The authors do qualify their proposal stating that the peptides “potentially promote a transition”. Nevertheless, it seems that the results would need to be augmented and further analyzed.

The authors propose a model of OMP insertion that adds a new state to the current consensus model. In this state, a nascent OPM beta-sheet appears to be inserted in the membrane with one edge engaged with BamA and the other open and exposed to the hydrophobic lipid core of the membrane. This would be quite unfavorable. Along the same lines, the often-hydrophilic residues normally facing the inside of a beta barrel would also be exposed to the hydrophobic core in this inserted beta sheet state. Even if the observation that the cyclic peptides accelerate the assembly of EspP(G1123R) can be confirmed, it seems that a more likely candidate for acceleration may be the release step (transition from “barrelization and strand exchange” to “closed BamA” in the authors sequence), although it is not obvious that this is the rate limiting step for EspP(G1123R) assembly.

Overall, the identification of inhibitors that can bind with high affinity to the lumen of BamA and allosterically inhibit OMP assembly is a significant finding that is very well supported in the manuscript. On the other hand, the data supporting the proposal of a folding acceleration effect of the cyclic peptides and the additional state in the mechanism of BamA requires additional evaluation and discussion.

Reviewer #5

(Remarks to the Author)

Version 1:

Reviewer comments:

Reviewer #1

(Remarks to the Author)

Dr. Walker, Dr. Bernstein, and their colleagues have provided satisfactory responses to all the reviewer questions. The changes and additions to the manuscript are all appropriate and improve the overall quality of this paper. This reviewer also appreciates the thoroughness of the responses and the authors’ willingness to provide additional insight and reflection on questions that were outside of the scope for this work. As previously noted, this remains a well-written, timely, and scientifically sound manuscript that advances the fields of outer membrane protein folding, antibiotic discovery, and bacterial physiology and is appropriate for the readership of Nature Communications.

I do have one additional minor suggestion. Now that the authors bring up the recent Genentech work describing distinct BamA-inhibiting macrocycles, it might be worth noting (as was noted in that Sun et al. manuscript) that such macrocycle discovery approaches can generate different peptides depending on the target presentation and panning conditions. This rationalizes the abilities of the two groups to discover distinct peptides with novel mechanisms and further highlights the strength of this approach to identify multiple, novel inhibitors for challenging drug targets. This is not necessary for publication or moving this manuscript forward, but just something for the authors to consider.

Reviewer #2

(Remarks to the Author)

The authors have thoroughly addressed the majority of the remarks we provided, offering well-articulated and detailed explanations throughout the revised manuscript. Their responses are thoughtful and effectively clarify the points raised during the review process.

One of the additions is the inclusion of a new scheme in Figure 6, which significantly enhances the reader's understanding of the experimental design and findings. Additionally, the authors introduced sequence alignment data in Figure S13, which highlights the peptide binding residues. This inclusion is particularly valuable, as it sheds light on the specificity of the peptides and supports the interpretation of the results. Moreover, the revised Figure S6 now includes data from a broader range of bacterial strains (*E. coli*, *Acinetobacter baumannii* ATCC17987, *Pseudomonas aeruginosa* MB5919, and *Staphylococcus aureus* Col). This expansion strengthens the manuscript by addressing the spectrum of bacterial activity and providing a more comprehensive analysis.

Regarding the solubility and cytotoxicity of the peptides, while direct assays were not conducted, the authors noted that the peptides were completely soluble at a concentration of 5 mg/mL, remained stable under these conditions, and exhibited no toxic effects on *E. coli*. Although these observations suggest favorable properties, specific assays to validate solubility, stability, and cytotoxicity would add robustness to these claims.

The authors also implemented several textual and structural improvements. They replaced “in vivo” with “in live cell” throughout the manuscript, as suggested, ensuring greater accuracy. The peptide JB-95 was included in the introduction, aligning with the feedback to provide a more comprehensive background. Additionally, they incorporated suggested references and expanded information in Table S5 and Figures S25–S27, which enhance the manuscript's thoroughness and overall quality.

In light of these substantial improvements, I recommend this manuscript for publication in Nature Communications. The revisions not only address the initial concerns but also elevate the clarity and scientific rigor of the work.

Reviewer #4

(Remarks to the Author)

The authors have satisfactorily addressed my previous comments

Reviewer #5

(Remarks to the Author)

REPLY TO REVIEWERS' COMMENTS (major changes are highlighted in the text)

Reviewer #1 (Remarks to the Author)

Folding and insertion of beta-barrel proteins into the gram-negative outer membrane is an essential process carried out by the beta-barrel assembly machine (BAM). BamA, the central and conserved component of the BAM complex, is a potential antibiotic target, however most recently described inhibitors have key limitations. In their manuscript Antibacterial macrocyclic peptides reveal a novel mode of BamA inhibition, Dr. Harris Bernstein, Dr. Scott Walker, and co-workers describe the discovery and characterization of three novel BamA-targeting macrocyclic peptides. Though these peptides fall short in their ability to inhibit growth of wild-type bacteria, they are enabling in that they identify a previously unappreciated inhibitory binding sites in the BamA lumen, display antibacterial activity against sensitized strains, and enable important characterization of outer membrane protein biogenesis. Overall, these peptides provide compelling insights that inform the mechanism of BAM and future antibiotic discovery and optimization.

The experiments presented in the paper are well-designed, thoroughly justified, and logical. The manuscript is very well written, and the conclusions drawn by the authors are justified by the clearly presented data. The exciting results presented in this work reveal important insights that will be of high interest to broad readership of Nature Communications. Though I have no major concerns about this manuscript in its current state, I have provided some questions and comments below that could help clarify some points or, in some cases, extend the findings.

- Was there a rationale for choosing the non-natural amino acids that were included in the NNU1 and NNU3 libraries you generated?

NNU1 and NNU3 are codon tables containing *N*-methylated amino acid and *N*-alkylated amino acids that have been optimized to translate with high efficiency and fidelity, allowing for robust production of macrocyclic peptide libraries. Historically, these two peptide libraries have been used in numerous in-house screening campaigns, resulting in the isolation of validated high-affinity binders. We have edited the text on p. 7 to reflect this more clearly: "Peptides of interest originated from two libraries named "NNU1" and "NNU3" that encode *N*-methylated and *N*-alkylated amino acids and that have been optimized to translate with high efficiency and fidelity."

- How many different peptides were encoded in the library after the 6th round of enrichment and what were the criteria used to choose CP1, CP2, and CP3 for follow up characterization? Were there any variants (e.g., substituted amino acids at single positions) of CP1, CP2, or CP3 present in the enriched libraries of BamA binders? This could provide extremely valuable insight for any future efforts to optimize these macrocycles.

Depending on the sequencing depth, the 6th round could still be composed of a diverse composition of peptides. To prioritize candidates (including CP1, CP2, and CP3), we only synthesized unique peptides that, by copy number after sequencing, represent at least 1% of the gene pool. These chemically synthesized peptides were then validated using SPR.

We did observe variants of CP1, CP2, and CP3 in our screen. We agree with the reviewer that these variants provide crucial information given that a high frequency of variation at any position suggests that that position is tolerant to mutations and could be a starting point for optimization. We would use a systematic method to optimize the peptides as well as consider the significance of the variants, but the peptides were not further optimized due to their lack of broad spectrum

activity.

- The binding spectrum of CP1, CP2, and CP3 provides useful information. Was binding to BamA from other Enterobacteriaceae species assessed (e.g., *Klebsiella pneumoniae*)? Are you able to rationalize the selective binding to *E. coli* BamA over *P. aeruginosa* BamA and *A. baumannii* BamA by the amino acid identities at the structurally determined macrocycle binding sites? This could point to interactions that could be the focus of future efforts to broaden their spectrum.

We agree with the reviewer that the binding spectrum of the peptides provides useful information. To rationalize the selective binding of the peptides to *E. coli* BamA we have added a multiple sequence alignment (Fig. S13) that highlights the peptide binding residues to clarify that they are poorly conserved across multiple species. We do not currently have purified *Klebsiella* BamA available for binding assays, but the overall sequence identity to *E. coli* BamA is very high (91.1%). Indeed, the sequence alignment shows that the binding residues are highly conserved within the Enterobacteriaceae, which suggests that the peptides would likely bind to the *Klebsiella* BamA isoform. We have edited the text on p.11 to clarify the lack of conservation among non-Enterobacteriaceae: "Interestingly, we found that while the BamA side chains that form key interactions with CP1-CP3 are highly conserved in the Enterobacteriaceae, they are much less conserved between *E. coli* and *A. baumannii* and *P. aeruginosa* (Fig. S13). The lack of sequence conservation likely explains why the peptides did not bind to the BamA isoforms produced by these organisms in our SPR experiments (Fig. S5).".

- Did you evaluate binding of CP1, CP2, and CP3 to the BAM complex (and not just BamA) by SPR? Given the fact that reported structures of BamA differ at the lateral gate based on the presence of the accessory lipoprotein, it could provide some insight into binding (and could explain the diminished activity you observe with the strains lacking BamB).

We did not evaluate the binding of the peptides to the BAM complex, and we agree with the reviewer that the presence of the lipoproteins might affect their binding to BamA. Because our crystal structures show that CP1 and CP3 bind across the lateral gate, however, it seems extremely unlikely that these peptides would be able to bind to BamA with high affinity in an open state. Indeed it was shown by Kaur H et al. Nature 2021 (ref. 44) that the binding of darobactin, which binds primarily to BamA β 1 but also crosses the lateral gate and interacts with BamA β 16, is not affected by the presence of the BAM lipoproteins. Given that the properties of all three cyclic peptides were very similar in all of our assays, it also seems unlikely that the presence of the BAM lipoproteins significantly affects the binding of CP2 to BamA or the state of the lateral gate after CP2 binds to BamA.

We also want to note that our analysis of the effect of the peptides on the growth of a BamB deletion strain (MC4100 Δ *bamB*) and an isogenic strain that contains the BamA E470K mutation was performed to obtain evidence that the cyclic peptides function by binding to BamA and not by acting as a non-specific toxin. Because Δ *bamB* strains (like *bamA101* strains) have compromised BAM activity, we would have expected that the growth of the Δ *bamB* strain we tested would be impaired by the peptides (as it was). The observation that the BamA E470K mutation suppresses the growth defect strongly suggests that the binding of the peptides to BamA (which, at least in the case of CP2 and CP3, would likely be affected by the mutation)—and not the absence of BamB—explains the results. We have now modified the text on p. 8 to clarify this line of reasoning.

- Have you tested the whole cell activity of CP1, CP2, and CP3 against *E. coli* strains with a compromised outer membrane barrier (e.g., imp4213 mutant) or in the presence of outer membrane permeabilizing agents (e.g., polymyxin B nonapeptide)? This could provide insight into the points raised about how these peptides access their target within the BamA lumen. You should also consider testing these against efflux defective strains.

There are some additional useful controls to add for the MIC testing. First, MICs against *P. aeruginosa* and *A. baumannii* as negative controls (you demonstrated that the macrocycles do not bind to their BamA). Second, an MIC against a gram-positive strain would identify any non-specific activity (or this could be assessed in a mammalian cell assay). Third, with wild-type *E. coli*, even though they do not kill the bacteria, it could be enabling to measure if macrocycle inhibition of BamA is sufficient to potentiate the activity of antibiotics that are normally excluded by the intact outer membrane. Finally, given the observations with darobactin, I am wondering if you considered looking at whether there is an additive, synergistic, or even antagonistic effect if you co-treat with darobactin and one of your macrocycles.

We agree that some of these controls are important and were actually part of our workflow, but we did not include the data in our original manuscript. The hits were initially screened for antibacterial activity against a wildtype *E. coli* strain, an *E. coli bamA101* strain, *E. coli* MB5746 (an *E. coli ΔtolC/lpxC* strain that has a relatively permeable outer membrane and a reduced drug efflux function), and a wildtype *Staphylococcus aureus* strain. Only hits that met the following criteria were selected for further profiling: 1) activity against *bamA101* strain > activity against wildtype strain, 2) activity against *bamA101* strain > activity against MB5746, and 3) no activity against *S. aureus*. The goal was to remove compounds whose activity was largely mediated by traversing the outer membrane instead of by engaging with BamA. Additional whole cell profiling showed that none of the peptides had any activity against *A. baumannii* and *P. aeruginosa*.

To address the reviewer's concerns we have now added new data that shows the antibacterial activity of the peptides against all of the above *E. coli* strains and other bacteria (Fig. S6) and edited the text on p. 8 to include the following: "To confirm that the antibacterial activity of CP1-CP3 against JCM972 was due to the binding of the peptides to BamA and not simply to the loss of membrane integrity that results from the *bamA101* and *ΔbamB* mutations, we also measured the activity of the peptides against *E. coli* MB5746, a *bamA+* strain that has a more permeable OM than JCM972 and that is defective in drug efflux. The peptides inhibited the growth of MB5746, but less effectively than the growth of JCM972 (Fig. S6). This observation indicates that the toxicity of the peptides does not solely correlate with the permeability of the OM and strongly suggests that they act by inhibiting BamA function directly. Finally, consistent with the binding results described above, we found that the peptides did not affect the growth of *A. baumannii* or *P. aeruginosa* or the Gram-positive organism *Staphylococcus aureus* (Fig. S6)." We also note on p. 14 that "the intermediate sensitivity of strain MB5746 to the peptides (Fig. S6) might be due to the permeability of its OM".

We also agree with the reviewer that it would be interesting to examine the antibacterial activity of darobactin and other compounds in the presence of our peptides, but we believe that these experiments are beyond the scope of the current study.

- Have you attempted to select for mutants resistant to CP1, CP2, or CP3? These selections, even with the sensitized strains, could be extremely informative to provide additional insight into binding, identify any possible bypass resistance mechanisms, further differentiate the three macrocycles, and illuminate BamA function.

We agree with the reviewer that the isolation of resistance mutants would be very informative, but we believe that such an endeavor is beyond the scope of the present study. We have already shown that BamA residue E470 interacts with CP2 and CP3 and that the E470K mutation suppresses the sensitivity of strain A710 to this peptide. Although we would presumably isolate mutations in other BamA residues that interact with the peptides in a resistance screen, we would likely also isolate mutations elsewhere in BamA and possibly other proteins that would be require a considerable amount of work to characterize.

- Not suggesting adding more structures (already a ton of useful information in the manuscript), but did you consider looking at the structures of BamA with just CP2? Also, how about the structures of any of the macrocycles with the full BAM complex instead of just BamA? I guess it is not entirely clear why your chose to solve the structures of BamA with CP1 and CP2 in the presence of darobactin and CP3 in the absence.

Before we started the present study, we had a well-established co-crystallization system with darobactin already bound to BamA. Initially we chose to soak CP1 and CP2 peptides into the preformed co-crystals, and because darobactin did not dissociate from BamA even though we soaked in the cyclic peptides over a period of 8-24 hours using a solution that lacked darobactin, we observed darobactin in our structures. We co-crystallized CP3 and BamA without darobactin to confirm that the presence of darobactin does not affect the binding of the cyclic peptides, and we found that the BamA β -barrel bound to CP3 maintains the same conformation as it does when it binds to both darobactin and CP1 or CP2. The structural data, along with the results of our SPR analysis (Fig. S4) suggest that the binding of darobactin to BamA does not significantly affect the binding of CP1 and CP2 to BamA. For this reason we do not believe that any additional insights would emerge from an analysis of the structure of CP1 and CP2 bound to BamA without darobactin.

It might be interesting to examine the structures of the macrocycles bound to the complete BAM complex, although we believe that the additional structural studies are beyond the scope of the present study. We would also like to mention that it has already been shown that the structure of darobactin bound to the complete BAM complex is essentially identical to its structure bound to BamA alone (ref. 44) even though darobactin binds to closer to the BamA barrel-POTRA domain interface than the macrocycles that we isolated.

- The in vivo folding assays are super informative. Can you speak to the degree of OMP folding inhibition and the impact on growth of *E. coli*? Presumably the lack of complete inhibition by the macrocycles accounts for their inability to kill wild-type *E. coli* but this is still sufficient to kill the sensitized strains. Does this provide insight on exactly how much BamA inhibition *E. coli* can tolerate?

As we show in Fig. S18, our macrocycles only inhibit OMP assembly in a wild-type strain by ~50%, and it has been known for many years that this level of inhibition is insufficient to kill *E. coli*. On the other hand, it would be very difficult to determine exactly how much BamA inhibition *E. coli* can tolerate from our data. This is an interesting question that is much more complicated than it might seem. It has been clear for a long time that growth conditions affect the degree to which *E. coli* can tolerate any kind of stress, so it is likely that the cells can tolerate more BamA inhibition if grown in minimal medium (which is a permissive growth condition) than if grown in LB. In addition, BamA inhibitors might affect the assembly of individual OMPs differently. As we show in Fig. 5, the addition of CP1-CP3 impairs the assembly of EspP Δ 5 and OmpC more than the assembly of OmpA. In principle, a given inhibitor might strongly reduce the assembly of

many non-essential OMPs but not affect the assembly of LptD, which is required for viability, and consequently produce only minor growth defects. Although it would be difficult to put a number on it, based on the literature it would be fair to conclude that *E. coli* can tolerate quite a bit of BamA inhibition (as well as defects in OM integrity).

- The analysis of the BamA lateral gate hole radius is quite interesting. Curious about whether you can model substrates in this as well? Also, it looks like darobactin was excluded in the MD simulations with CP1 and CP2 (Fig. 3A and 3B). Is this correct? Would you predict that the presence of darobactin would have any impact? Visually, I found it a little difficult to compare across the traces for the different replicates and conditions (for example in Fig. 3c and 3d) and thought it might be helpful to have some overlay versions (even if only in the supplemental data) to better illustrate the comparisons.

The presence of substrates would not make an impact on the hole calculation itself, which involves moving a cone or cylinder into the protein cavity to assess volume/space. The presence of a substrate might certainly affect the results of the MD simulations, but we believe that such an experiment is outside of the scope of this work. Although the presence of darobactin would likely have an impact on the end result, the purpose of the MD simulations was to assess the channel and gating dynamics of BamA in the presence of two novel macrocyclic peptides (CP1 and CP2). We removed darobactin to focus on CP1 and CP2.

As suggested by the reviewer, we have modified Figs. 3C-D and S24 by adding crosshairs that make it easier to compare traces from each of the replicates at a Z coordinate of 14 Å. We believe that this representation will be more helpful than an overlay.

- Regarding the 'inaccurate initial docking position of the peptide' (CP1 hole radii experiments at the bottom of page 16), how many different starting positions did you explore? Could other starting positions provide a different picture of what is going on here?

The initial placement of the ligand was performed by overlaying the structure of BamA bound to CP1 in the closed state to the apo structure of BamA in the open state. The ligand was then relaxed to optimize the geometry in the proposed binding site in the closed state. We have now added these details to the Methods section and modified the text on p. 17 to indicate that "The instability is likely attributable to an inaccurate initial docking position of the peptide within BamA in its open state. Indeed because the crystal structure shows CP1 bound across the lateral gate of BamA in its closed state (Fig. 2B), it is difficult to dock the peptide into the open state structure." As the reviewer suggests, different starting positions might provide a different picture, but would require performing many more docking trials and MD simulations and possibly obtaining additional crystal structures. We believe, however, that a full analysis of the effect of CP1 on gating is beyond the scope of the present study.

Minor:

- In Fig. 1B, the macrocyclization bond appears to be off in CP2 and CP3 cartoons (at least in the printout I made).

Thanks for pointing out the error. We have now corrected the cartoons.

- On page 16, for the CP1 hole radii, it refers to Fig 3C and S22, but I don't think the CP1 results

are in 3C (unless it is referring to 3C for the apo...in which case, can you replicate the apo data in S22 to allow for an easier direct comparison).

As suggested, we have amended Fig. S22 (now Fig. S24) to replicate the apo data from Fig. 3C for clarity.

- What is the blue structure in Fig. S10A (I think CP1 is pink and BamA is grey, but completely missed what the blue structure is)?

The blue structure is darobactin. We have modified the figure legend to clarify the identity of this structure.

Reviewer #2 (Remarks to the Author)

Antibacterial macrocyclic peptides reveal a novel mode of BamA inhibition

Summary/key results

The manuscript focuses on the discovery of novel cyclic peptides that inhibit the essential β -barrel assembly machinery (BAM) in *E. coli*, specifically targeting its central component, BamA. BAM is responsible for the folding and membrane insertion of outer membrane proteins (OMPs), and BamA is a key antibacterial target in Gram-negative bacteria due to its conserved structure and critical function. Using mRNA display, the researchers identified three monocyclic peptides (CP1-3), from a library of 1012 unique cyclic peptides, that bind with high affinity to BamA, inhibiting bacterial growth in *E. coli* strains with BAM deficiencies, and OMP assembly of several model *E. coli* proteins both in vitro and in vivo. Remarkably, unlike previously characterized inhibitors, such as darobactin, these cyclic peptides bind to unique sites within the BamA β -barrel lumen. The crystal structures show that CP1 and CP3 lock the gate by binding to both sides of the BamA β -barrel, while CP2 restricts its flexibility. MD simulations confirmed the stable binding of these peptides. Taken together, these results revealed a novel mechanism of inhibition. Intriguingly, these peptides also accelerate the assembly of a mutant OMP when added after it binds to BamA, suggesting that the peptides stabilize BamA in a conformation that blocks early stages of OMP assembly but facilitates later steps. This work highlights a new potential approach for antibacterial drug development by inhibiting BamA to control OMP assembly.

Validity

Good.

The work presented is well executed and will be of interest to readers of Nature Communications. Before publication, a few minor formalities need to be addressed.

The interpretation of the data in this study appears valid and is supported by a series of complementary experiments (binding assays, MIC assays, crystal structures, MD simulations and assays to evaluate the inhibition of OMP assembly in vitro and in vivo). The conclusions regarding the novel mode of inhibition of BAM by cyclic peptides are well-founded, as they are consistently supported by the experimental data. However, although the results are convincing, the high concentrations of peptides required for activity could limit their immediate therapeutic applicability, which the authors acknowledge, and it is not the main focus of the study. Further exploration of the specificity of peptides for BamA, such as proteomic studies, would strengthen and broaden the impact of the study's findings.

This new approach could also inspire further exploration of cyclic peptides or small molecules

that target essential bacterial pathways by similar mechanisms.

Significance

High

This manuscript presents the discovery of a new mode of inhibition of BamA, a promising target for the development of new therapeutic strategies against Gram-negative bacterial infections. By identifying cyclic peptides that bind to BamA, the study reveals a new mechanism distinct from known inhibitors such as darobactin. These peptides uniquely interact with sites in the β -barrel lumen of BamA, blocking early steps in outer membrane protein (OMP) assembly while accelerating later steps under certain conditions. Although high peptide concentrations are required for activity, the research focuses on the novel binding mechanism. The discovery made by the authors is important and will have consequences in the field of BamA inhibitors.

Data and methodology

Good.

The manuscript is well-structured, with a logical progression from background to results and conclusions. The use of the mRNA display platform to generate a large peptide library is well justified, focusing on BamA in *E. coli* to target a key bacterial pathway. While the antibacterial spectrum is narrower compared to darobactin, the study introduces a novel mode of BAM inhibition, adding a unique aspect to the research. The approach is solid and the assays that were chosen for this article to prove a new mechanism of interaction with BamA are relevant. Figures are clear and informative. The data presented are of good quality, with sufficient controls (darobactin, just DMSO...). However, there is no mention of how many replicates were performed for each experiment (Figures 4, 5 and 6). Add the gels and data related to the replicates in the SI. Moreover, additional dose-response data would be useful to better assess the concentration-dependent effects of peptides in various assays, especially for the assay Figure 4F (1 or 2 points after 2 μ M).

We already note that the experiments shown in Figs. 4 and 6 were performed three times. We have now performed all of the experiments shown in Fig. 5 three times (and repeated the related experiment in Fig. S18) and have modified the Figure legend to indicate the number of replicates. While we agree that additional dose-response data would be useful in Fig. 4F, one can easily determine the concentration at which a 50% reduction in assembly was observed from the data that is already provided. Furthermore, to obtain additional dose-response data we would need to purify more BAM, reconstitute the complex into proteoliposomes, and perform the experiments shown in Fig. 4F three more times. This would be a time-consuming process that would not significantly enhance our ability to draw solid conclusions from the data.

We have placed all of the gels that are not shown in Figs. 4-6 in the Source File.

The subsequent analysis of the results is generally good but could benefit from additional clarification and a more balanced interpretation.

Please find below my remarks and questions for the different parts of the paper.

Abstract

Page 1. You wrote: “. We identified peptides that arrest *E. coli* growth,”. However, the cyclic peptides were antibacterial when added to *E. coli* strains that have BAM deficiencies. Please rephrase.

As suggested, we rephrased the sentence to read “We identified peptides that arrest the growth of BAM deficient *E. coli* strains...”

Page 1. Please replace “in vivo and in vitro” with “in vitro and in vivo” and check the usage of “in vivo” vs “whole-cell” (c.f. comment to page 8)

As the reviewer suggests, we have changed “in vivo” to “in live cells” throughout the text.

It may be worth including a short explanation on the source of the cyclic peptides in the abstract.

We agree that the discovery method is a crucial part of our study, but due to space limitations we cannot describe the method in the Abstract. We do, however, describe all of the important details of our mRNA display screen in the Results section.

Introduction

- Page 4. You describe briefly some known BamA inhibitors. Please add other cyclic peptides obtained by synthesis, such as JN-95. This part could be described in more detail to show that there are not many effective darobactin inhibitors already described, particularly peptide inhibitors.

As suggested by the reviewer, we now discuss JB-95 in the Introduction and cite the appropriate reference (p. 4). For readers who are interested in other potential BamA inhibitors, we also cite another recent review (ref. 35). Finally, we discuss two new peptidic BamA inhibitors that were isolated by Genentech by mRNA display and reported after we submitted our manuscript (p. 5)

- The darobactin is a bicyclic molecule, so why did you decide to focus only on monocyclic synthesis? It would be also interesting to have bicyclic peptides in this study to mimic darobactin. Page 5 : “Given that darobactin is based on a bicyclic peptide scaffold, we reasoned that small, constrained cyclic peptides might be the most suitable modality for BamA inhibition.” Can your peptides be described as small and constrained? A related question is whether it should be made more explicit that the cyclic peptides were initially designed to mimic darobactin. As a result, the new binding mode is surprising.

Although we agree with the reviewer that it would be interesting to screen bicyclic peptides that mimic darobactins in the future, we were not really attempting to mimic darobactins in this study. Our main goal was to exploit the finding that BamA is amenable to inhibition by peptidomimetics, including darobactins. To clarify our reasoning, we now indicate that our use of mRNA display was based partly on the finding that peptidomimetics other than darobactins bind to BamA even if their ability to inhibit BamA activity has not yet been determined (p. 5, top).

We certainly consider the library of cyclic peptides that we screened to be constrained (unlike linear peptides). In contrast, we have removed the word “small” because our cyclic peptides are 15-16mers that are much larger than darobactins.

Results and discussion

- Page 6. Were only 3 peptides from the large library shown to bind the *E. coli* BamA β -barrel in vitro? If not, could you briefly explain how these 3 peptides were selected?

We indeed identified other promising macrocyclic peptides that bind to the *E. coli* BamA β -barrel with high affinity, but as we now state on p. 7, we selected CP1, CP2, and CP3 for further study based partly on the results of preliminary growth inhibition experiments. We first noticed that these peptides function similarly in cell-based assays (i.e., they only inhibit the growth of *E. coli*).

We then found that they all bind to the lumen of the BamA β barrel, which distinguished them from all of the BamA inhibitors that had been described when we started our study.

- Page 7-Figure 1. The MIC was determined only on *E. coli*. Why did you choose only 3 Gram-negative strains for BamA binding? Given that they bind only to *E. coli* BamA, tests on other strains, like the strains mentioned in the introduction (*K. pneumoniae*, *Enterobacter* spp) would be interesting? BamA binders are supposed to be active on only Gram-negative bacteria. As a result, it would be important to confirm the lack of activity against Gram-positive bacteria (as a negative control because Gram-positive do not possess the BAM complex and they are not affected by the darobactins). Indeed, on page 4 in the introduction, a drawback of MRL-494 is mentioned as it exhibits off-target activity against Gram-positive bacteria.

As we mentioned in our reply to reviewer 1, the hits were initially screened for antibacterial activity against a variety of *E. coli* strain and a wildtype *Staphylococcus aureus* strain. Only hits that showed no activity against *S. aureus* (and met other criteria) were used for further profiling. We now include evidence that CP1-3 did not affect the growth of this Gram-positive organism in Fig. S6. Additional whole cell profiling showed that none of the peptides had any activity against *A. baumannii* and *P. aeruginosa* (also now shown in Fig. S6).

To address the specificity of the peptides, we have now included a sequence alignment in Fig. S13 that shows that many of the key residues in BamA that form contacts with CP1-3 are conserved in several species that are closely related to *E. coli* (*Klebsiella*, *Citrobacter*, *Enterobacter* and *Salmonella* species), but are not conserved in *A. baumannii* and *P. aeruginosa*. We also note the different levels of sequence conservation on p. 11. Although we have not tested the activity of the peptides against other Enterobacteriaceae, based on the sequence alignment it seems likely that they can bind to the BamA homologs produced by those organisms with a reasonable affinity.

- Page 7. The peptides interact with BamA but the MIC values on *E. coli* WT are very high. Can you explain that? Do you think that the different position in the BamA pocket can explain the high MIC values?

We hypothesize that the MIC values are high because the peptides cannot readily cross the outer membrane. This hypothesis is corroborated by our *in vitro* data, which demonstrate that the peptides are much more active in a system in which they are exposed to the periplasmic side of BamA instead of the extracellular side. We present this idea in the Discussion (p. 18): “Interestingly, we found that although the cyclic peptides all bind stably to distinct sites in the lumen of the BamA β -barrel, unlike darobactin, they inhibit BAM activity at a significantly lower concentration when they have direct access to the periplasmic side of the BamA β -barrel. This result together with the high MIC values for wild-type *E. coli* strongly suggests that the cyclic peptides either enter the BamA lumen from the periplasmic side but cross the OM poorly because they are too large and hydrophobic to be transported effectively through porins, or that they can enter the BamA lumen from the extracellular side, but relatively inefficiently.”

Please Add the MIC values on AM710 (MC4100 Δ bamB36) and AM711, on Figure 1D.

We did not determine the MIC values for these two strains because they were tested solely on agar plates as a control to provide further evidence that the peptides act by inhibiting the function of BAM.

- Page 7. Did you test the affinity of the peptides to LptD which is also an OMP and essential for viability, and more largely different transmembrane proteins? For example the peptide JB-95 targets BamA and LptD (Urfer, M.; et al., A peptidomimetic antibiotic targets outer membrane proteins and disrupts selectively the outer membrane in *Escherichia coli*. *J. Biol. Chem.* 2016, 291, 1921–1932.)

We did not test the affinity of the peptides for LptD or other outer membrane proteins given that our SPR experiments provide strong evidence that CP1-CP3 bind selectively to *E. coli* BamA, and do not even bind to other BamA isoforms. The selectivity of the peptides was also validated by our crystal structures. Based on our results, we have no reason to suspect that the peptides bind to any off-target proteins. Even if the peptides bind to other outer membrane proteins with low affinity, the finding that the peptides impair the growth of BAM-deficient cells more than strain MB5746, which phenocopies LptD-deficient cells by producing a reduced level of LPS, indicates that the major target of CP1-CP3 is BamA.

We should also note that JB-95, which was not selected based on its binding to BamA, binds to multiple outer membrane proteins including LamB and FadL, and also permeabilizes both *E. coli* cell membranes (ref. 42).

- Page 7. You wrote “. The activity of the peptides against these mutant strains suggests that they not only bind to BamA, corroborating the SPR binding data, but also impair its function.” Did you consider to perform proteomics studies, like Photo-affinity interaction mapping, to evaluate the impact of these new peptides on proteins?

We agree that further assessing the influence of our peptides (and other BamA inhibitors) on overall protein levels in the cell would be very informative, but such experiments are beyond the scope of the current work.

- Page 7. You wrote “Interestingly, the peptides did not bind to either *P. aeruginosa* or *A. baumannii* BamA β -barrels (Figure S5) and therefore lack the broad-spectrum activity of darobactin.” Please provide a possible explanation?

As noted above, we have now included a sequence alignment (Fig. S13) that shows that many of the key residues in BamA that form interactions with the peptides are not conserved in *p. aeruginosa* and *A. baumannii*. As we note on p. 11, “the lack of sequence conservation likely explains why the peptides did not bind to the BamA isoforms produced by these organisms in our SPR experiments”.

- Page. 8: “and as it is when darobactin binds to BamA in vivo⁵⁵.” I would recommend to carefully review the use of the term “in vivo” as it seems to be always used when whole-cell experiments with bacteria are meant. In drug discovery, “in vivo” is normally reserved for experiments in animal models. This may confuse the readers.

As noted above, we have now replaced “in vivo” with “in live cells” throughout the text.

- Page 8. “Because darobactin does not affect the ability of the cyclic peptides to bind tightly to BamA”. It may be more prudent to write: darobactin does not appear to affect the ability to...

We have added “appear to” the text as suggested.

- Page 8. For the crystallisation experiments with CP1 and CP2, BamA was used in complex with darobactin for soaking. Why was the same protocol not also used for CP3?

As we mentioned in our reply to reviewer 1, before we started the present study we had a well-established co-crystallization system with darobactin already bound to BamA. We chose to soak CP1 and CP2 peptides into the preformed co-crystals, and because darobactin did not dissociate from BamA even though we soaked in the cyclic peptides over a period of 8-24 hours using a solution that lacked darobactin, we observed darobactin in our structures. We co-crystallized CP3 and BamA without darobactin to confirm that the presence of darobactin does not affect the binding of the cyclic peptides, and we found that the BamA β -barrel bound to CP3 maintains the same conformation as it does when it binds to both darobactin and CP1 or CP2. The structural data, along with the results of our SPR analysis (Fig. S4) suggest that the binding of darobactin to BamA does not significantly affect the binding of CP1 and CP2 to BamA. For this reason we do not believe that any additional insights would emerge from an analysis of the structure of CP1 and CP2 bound to BamA without darobactin.

- Page 10. “Cyclic peptides identified by mRNA display strongly inhibit BAM activity in vitro and in vivo”. They did not strongly inhibit BAM activity in vivo, please change the phrasing and also reconsider the use of “in vivo”. Moreover, you wrote page 12 “Consistent with our in vitro data, we found that CP1, CP2 and CP3 also inhibit BAM activity in vivo, but less effectively.”

As suggested, we have removed the word “strongly” and changed “in vivo” to “in live cells”

- Page 12. For the experiment with living bacteria, please confirm whether the solubility, cytotoxicity and the stability of the peptides was determined before running these studies.

We did not conduct a formal analysis of the solubility or stability of the peptides under different experimental conditions. We dissolved the peptides in DMSO and found that they were completely soluble at a concentration of 5 mg/ml. As we now state in the Methods section (p. 31), we diluted the peptides in DMSO and added them directly to the agar used for the whole cell inhibition assays. The peptides clearly remained stable in solution long enough to produce the growth defects shown in Fig. 1E. In pulse-chase assays, the solubilized peptides were also added at the appropriate concentration directly to cultures. Once again, the peptides must have remained stable in solution for 30 min to generate the results shown in Fig. 6A. If the peptides were insoluble the strong inhibition of assembly observed when they were added before the chase (lanes 6-10) and the acceleration of assembly observed when they were added after the start of the chase (lanes 11-15) would not have been seen, or if they were unstable these effects would have dissipated at some point during the time course.

With respect to cytotoxicity, we found that the peptides do not kill wild type *E. coli* (Fig. 1D-E), but only effectively target *E. coli* strains with compromised BAM complexes or outer membranes. Likewise, the peptides inhibit the assembly of outer membrane proteins in a *bamA101* strain more effectively than in a *bamA+* strain (Figs. 5 and S18). These results strongly suggest that the peptides inhibit cell growth/outer membrane protein assembly through by engaging BamA and not by producing a non-specific cytotoxic effect.

- Page 13 and discussion part. Could an experiment be designed and run to validate the hypothesis concerning the difference in BAM inhibition in vitro vs in vivo?

That’s a great question. In principle it would be possible to design a genetic screen that selects for mutations that increase the sensitivity of *E. coli* to the cyclic peptides. Even if the desired

mutant strains could be isolated, however, it would be challenging to prove that the mutations increase the permeability of the outer membrane to the peptides or increase the entry of the peptides into the lumen of the BamA β -barrel from the extracellular side. We would need to generate radioactive forms of the peptides to analyze their localization and hope that we would see a difference in radioactivity between wild-type cells and hypersensitive cells (we could also generate fluorescent versions of the peptides, but then we would have to prove that the fluorescent groups do not interfere with their binding to BamA or their inhibition of BamA activity). In short, a considerable amount of work would be required to test our hypothesis, and we would have to be lucky enough to avoid potential technical problems that would undermine our efforts. For these reasons we believe that although it would be nice to prove our idea, the necessary experiments are beyond the scope of the present study.

SI

The SI is well documented, with figures and tables that support and reinforce the data presented in the paper. However, we do not have any information regarding the analysis of these 3 peptides. Please can you add at least the yield, HRMS and the purity of these compounds (chromatogram or elemental analysis).

This information is provided in Table S5 and Figs. S25-27.

Figure S19: please add the time of irradiation by UV.

This information is provided in the Methods section (p. 38).

Suggested improvements

See data and methodology for my remarks/questions.

Clarity and context

Overall, the manuscript is clear and well-structured, with research objectives and results presented in a logical manner. The introduction provides sufficient context to understand the importance of BamA as a target for Gram-negative bacterial infections and the use of cyclic peptides for antibacterial purposes. While the technical content is clear for an expert audience, certain sections (particularly the mechanistic details of OMP assembly) could benefit an additional explanation to improve accessibility to a broader readership, and perhaps need a scheme to aid understanding of the experiment.

To improve the accessibility of the technical aspects of the manuscript to a broad audience we modified the Results section and Figs. 5 and 6. First, we added a more detailed explanation of our pulse-chase experiments and clearly defined the forms of one of our model OMPs (EspP Δ 5) on p. 13. Second, we modified Figs. 5 and 6 so that the labeling is consistent. In both Figures we changed " β domain" to " β -barrel" to improve clarity and to maintain consistency with the text. To improve clarity we also changed "% cleaved" in Fig. 6 to "% proEspP(G1123R) cleaved". We realize that the experiments shown in Fig. 6 are somewhat complicated, and so to aid non-specialists we have added a schematic to Fig. 6 (Fig. 6A) that indicates the times at which various compounds are added and times at which important molecular events occur. To guide readers, we also cite Fig. 6A at appropriate locations on p. 15.

References

The manuscript appears to be well-referenced, with appropriate references to foundational and recent studies in the field of OMP assembly, BAM inhibition, and BamA inhibitors. Nevertheless, if there are any emerging studies or reviews that provide more comprehensive insights into

mRNA display techniques or recent developments in antimicrobial peptide research, including them would further strengthen the context.

Finally, you can add the review “Exploring the Darobactin Class of Antibiotics: A Comprehensive Review from Discovery to Recent Advancements”, ACS Infect. Dis. 2024, 10, 2584-2599”

To address this concern we now cite the suggested review article (ref. 35) and a new paper that was published after we submitted our manuscript in which the authors used mRNA display to discover novel BamA peptide inhibitors (ref. 54).

Your expertise

The last part of the paper, on the study of how peptides accelerate a later stage assembly, is outside of our expertise. This part was difficult to understand and may need a scheme to improve comprehension of the experiment.

As mentioned above, we have included a schematic in Fig. 6 (Fig. 6A) and modified the Y axis legends for the graphs in an effort to improve the comprehensibility of the experiments. To direct the reader to the right place, we have changed our initial reference to Fig. 6B from “Figure 6B, compare lanes 1-5 to 11-15 and curves on the right” to “Figure 6B, compare lanes 1-5 to 11-15 and the two curves in the graphs on the right”. We hope that irrespective of their expertise, readers will realize that the conclusion that the peptides accelerate a late stage of assembly is based on the clear differences between the red and blue curves.

Reviewer #4 (Remarks to the Author)

In this manuscript, Walker et al. implement an mRNA display approach to successfully select for cyclic peptides that selectively bind with low nanomolar affinity to E. coli BamA, the central component of the essential beta-barrel assembly machine (BAM). Unfortunately, the peptides did not display significant antibacterial activity against wild type E. coli, but the authors do show activity against E. coli mutants with impaired BAM function at micromolar concentrations. They further show that a mutant of BamA (E470K) suppressed the antibacterial activity of two of the peptides, validating BamA as the in vivo target for activity.

The cyclic peptides do not compete with darobactin, a competitive inhibitor of BAM that binds to the beta1 strand of BamA, suggesting a different mechanism of action. This is validated crystallographically, revealing BamA intra-lumen binding sites for all three peptides and providing a rationale for the suppression effect of the BamA E470K mutation. Furthermore, the authors clearly demonstrate BAM inhibition by all three target peptides both in vitro and in vivo using multiple model proteins.

Taken together, excellent data very strong support the conclusion that the selected cyclic peptides inhibit BAM-mediated OMP assembly by an “allosteric” mechanism: binding to the lumen of the BamA barrel that disfavor opening of the “lateral gate” required for full engagement of the substrate OMP. For CP1 and CP3 this appears to be due to binding across the gate, whereas CP2 may constrain the flexibility required for gate opening. Given the luminal binding sites for the peptides, I agree with the author’s interpretation that the low antibacterial activity despite the high binding affinity likely reflects poor periplasmic availability or impaired access to the lumen through the extracellular loops of BamA.

-The authors then go on to study the effect of the cyclic peptides in the in vivo assembly of the

slow folding mutant EspP(G1123R). Complete folding and assembly of EspP result in cleavage and release of the beta domain. They show that, as expected, addition of the peptides to the cultures before the chase step result in assembly inhibition. However, the observed effect of addition of peptides 2 minutes after the chase is interpreted as evidence of assembly acceleration. This interpretation is predicated on the quantification of the beta-domain bands as well as the preproEspP(G1123R) and proEspP(G1123R) bands. The authors then report the “percentage cleaved”, apparently normalized to the total for each time point. From visual evaluation of the gels, it seems that the main difference between the controls and the “2’ post-chase” conditions is the reduction in intensity of the ProEspP(G1123R) band, not increase of the beta domain. This could be due to faster degradation of the preEspP(G1123R) precursor in the presence of the peptides, particularly if there is a fraction that never folds. It is likely that addition of 25 μ M peptides would trigger the stress response with upregulation of protease activity in the periplasm, which may be responsible for the degradation of preEspP(G1123R). As the experiment seems to be evaluating the BAM activity under single turnover conditions for the labeled protein, quantification should be done normalizing the amounts of beta domain at each time point to the time zero amounts of preproEspP(G1123R) and proEspP(G1123R) bands. The time zero is identical for the control and the “2’ post-chase” condition and therefore appropriate for normalization. The authors observations may still hold true, but the suggested normalization would account for the possibility of degradation of a non-folding (or perhaps released from BAM) EspP(G1123R) fraction.

We agree with the reviewer that the stability of the EspP (G1123R) mutant is an important consideration. For that reason we compared the stability of the mutant protein in the absence or presence of the cyclic peptides, EDTA and sorbitol in Figs. S20, S22 and S23. The results show that, if anything, the mutant protein is *more* stable in the presence of a cyclic peptide or other compound that affects the properties of the outer membrane than in its absence. This observation is consistent with the idea that all of these compounds accelerate assembly and thereby expose the mutant protein to periplasmic proteases for a shorter period of time. If you carefully compare lanes 1-5 to lanes 11-15 in Fig. 6B or the first five lanes to the last five lanes in Fig. 6C and 6E, you should be able to see that the cleaved β domain accumulates faster while the pro form of the protein disappears more rapidly in the presence of a compound than in its absence.

In contrast, if you look at Fig. 6B, lanes 6-10, you should notice that there is only a small amount of the mutant protein that remains at the 30 minute timepoint because when the cyclic peptides are added before the protein is synthesized it is rapidly degraded by periplasmic proteases.

To aid the reader, we have now modified the legends to Figs. S20, S22 and S23 to explain in more detail how we calculated the stability of full-length EspP or the G1123R mutant.

-The evaluation of molecular dynamics experiments where CP2 was docked to the open conformation of BamA and compared to the apo-open and apo-closed conformations has several issues to be addressed. 50ns may be too short a simulation to study the desired effects. More than one replica of the apo-open trajectory should be considered and the closed state should also be relaxed to see if the system simply drifts to similar final state regardless of the starting apo state. Crucially, there is no statistical test to evaluate whether the differences are significant. The authors do qualify their proposal stating that the peptides “potentially promote a transition”. Nevertheless, it seems that the results would need to be augmented and further analyzed.

We agree that 50 ns is too short a time frame to assess the full conformational transition between the closed and open states; indeed, 500 ns or 5 μ s may not even be sufficient (Wu, R., Bakelar, J.W., Lundquist, K. et al. Plasticity within the barrel domain of BamA mediates a hybrid-barrel mechanism by BAM. *Nat Commun* 2021, 12, 7131. doi: 10.1038/s41467-021-27449-4). Such a transition would be very rare during standard MD simulations, and would likely require demonstrably converged biased MD, or other specialized techniques like Markov state modeling (Chodera JD, Noé F. Markov state models of biomolecular conformational dynamics. *Curr Opin Struct Biol.* 2014, 25:135-44. doi: 10.1016/j.sbi.2014.04.002) to provide enough observed open/closed transitions for a statistical analysis. For this reason, we did not anticipate observing any transition using this MD experimental design, and did not analyze for this in our MD simulations. We devised the molecular modeling and further MD simulation set-up to study the local dynamics of the channel in its open form or closed form independently, while bound to two novel macrocyclic peptides. The goal of such experiments was to illustrate any conformational dynamics induced by the peptides that could precipitate or pre-empt a transition, not to show a complete conformational transition from a fully open to fully closed state. For such local dynamics observations, a shorter timeframe such as 50-100 ns should be sufficient (Farmer J, et al. Statistical Measures to Quantify Similarity between Molecular Dynamics Simulation Trajectories. *Entropy* 2017, 19(12):646. doi: 10.3390/e19120646). We have edited the text to soften the language around “open” and “closed” conformational transitions to make this point clearer to the reader.

We performed three replicates of the MD simulations with peptides bound because this was a peptide-docked system and we wanted to mitigate the uncertainty of the docked peptide poses. We only performed one replicate of the apo state since it started from an experimental structure and was used only as a benchmark.

The closed state system was relaxed after docking the peptide, and we have modified the methods to clarify this point.

Our MD simulation trajectories captured the dynamics of the system, providing many snapshots (in this case, 2500) over the course of the simulation, in contrast to crystal structures which represent only a static single snapshot of the complex conformation. We have not used a statistical method to quantitate our results, because our interpretation of our simulation analysis remains qualitative only. A more statistically quantitative simulation to assess transitions between open and closed states of this protein is beyond the scope of the work described here.

-The authors propose a model of OMP insertion that adds a new state to the current consensus model. In this state, a nascent OPM beta-sheet appears to be inserted in the membrane with one edge engaged with BamA and the other open and exposed to the hydrophobic lipid core of the membrane. This would be quite unfavorable. Along the same lines, the often-hydrophilic residues normally facing the inside of a beta barrel would also be exposed to the hydrophobic core in this inserted beta sheet state. Even if the observation that the cyclic peptides accelerate the assembly of EspP(G1123R) can be confirmed, it seems that a more likely candidate for acceleration may be the release step (transition from “barrelization and strand exchange” to “closed BamA” in the authors sequence), although it is not obvious that this is the rate limiting step for EspP(G1123R) assembly.

Overall, the identification of inhibitors that can bind with high affinity to the lumen of BamA and allosterically inhibit OMP assembly is a significant finding that is very well supported in the manuscript. On the other hand, the data supporting the proposal of a folding acceleration effect

of the cyclic peptides and the additional state in the mechanism of BamA requires additional evaluation and discussion.

The first stages of our model are based entirely on previously reported results. As we note in the introduction, a previous cryo-EM analysis (ref. 25) provided strong evidence that OMPs begin to enter the OM as a curved β -sheet. Although in principle the insertion of an open (or partially open) β -sheet into a biological membrane would indeed be very unfavorable, the same cryo-EM study showed extreme thinning and disorder of the OM near the C-terminus of BamA. The magnitude of this thinning might drive the formation of a curved high-energy lipid structure around the BamA C-terminus that reorients the lipids in the space between BamA and the substrate so that polar lipid head groups transiently face polar side chains in the substrate. Although this is only one possible explanation of the available data, to address the reviewer's concern we have modified the legend to Fig. 7 to indicate that conformational changes in BamA during the early stages of the assembly process perturb the lipid bilayer and thereby might promote the membrane integration of OMPs as well as later stages of assembly.

We believe that our original results show clearly that the cyclic peptides accelerate the assembly of the EspP (G1123R) mutant. Furthermore, we have now added a new experiment (Fig. S19) that strongly supports the idea that the rate limiting step is the insertion of the mutant β -barrel into the OM. If, for example, barrelization were the rate limiting step, we would not observe the persistence of the ~26-28 kDa and ~12-14 kDa C-terminal fragments in the presence of PK that represent partially integrated EspP β -barrels. On the other hand, we completely agree with the reviewer that we should reevaluate our model. Our concern is that we might be overcounting the number of steps in the assembly process. Membrane insertion might not be a new step, as we originally proposed, but rather insertion, the formation of a hybrid barrel, and barrelization might all occur in one concerted step, so that by accelerating membrane integration the peptides might also accelerate the formation of a hybrid barrel and barrelization. We have now modified the Discussion (p. 21) and the legend to Fig. 7 to consider this possibility. We have also modified the last sentence of the Introduction, which now indicates that our results "*potentially* reveals a previously unidentified step in the OMP assembly pathway."

Reviewer #5 (Remarks to the Author)

REPLY TO REVIEWER'S COMMENT:

Reviewer #1 (Remarks to the Author):

Dr. Walker, Dr. Bernstein, and their colleagues have provided satisfactory responses to all the reviewer questions. The changes and additions to the manuscript are all appropriate and improve the overall quality of this paper. This reviewer also appreciates the thoroughness of the responses and the authors' willingness to provide additional insight and reflection on questions that were outside of the scope for this work. As previously noted, this remains a well-written, timely, and scientifically sound manuscript that advances the fields of outer membrane protein folding, antibiotic discovery, and bacterial physiology and is appropriate for the readership of Nature Communications.

I do have one additional minor suggestion. Now that the authors bring up the recent Genentech work describing distinct BamA-inhibiting macrocycles, it might be worth noting (as was noted in that Sun et al. manuscript) that such macrocycle discovery approaches can generate different peptides depending on the target presentation and panning conditions. This rationalizes the abilities of the two groups to discover distinct peptides with novel mechanisms and further highlights the strength of this approach to identify multiple, novel inhibitors for challenging drug targets. This is not necessary for publication or moving this manuscript forward, but just something for the authors to consider.

We thank the reviewer for this excellent suggestion! We have now added a sentence at the very end of the Discussion to note the power of mRNA display to identify multiple peptides that inhibit the activity of challenging drug targets by different mechanisms.